# Theoretical Foundations of Deep Selective State-Space Models

**Nicola Muca Cirone**
Department of Mathematics
Imperial College London

**Antonio Orvieto**
MPI for Intelligent Systems,
Tübingen AI Center
ELLIS Institute Tübingen

**Benjamin Walker**
Mathematical Institute
University of Oxford

**Cristopher Salvi**
Department of Mathematics
Imperial College London

**Terry Lyons**
Mathematical Institute
University of Oxford

## Abstract

Structured state-space models (SSMs) are gaining popularity as effective foundational architectures for sequential data, demonstrating outstanding performance across a diverse set of domains alongside desirable scalability properties. Recent developments show that if the linear recurrence powering SSMs allows for a selectivity mechanism leveraging multiplicative interactions between inputs and hidden states (e.g. Mamba, GLA, Hawk/Griffin, HGRN2), then the resulting architecture can surpass attention-powered foundation models trained on text in both accuracy and efficiency, at scales of billion parameters. In this paper, we give theoretical grounding to the selectivity mechanism, often linked to in-context learning, using tools from Rough Path Theory. We provide a framework for the theoretical analysis of generalized selective SSMs, fully characterizing their expressive power and identifying the gating mechanism as the crucial architectural choice. Our analysis provides a closed-form description of the expressive powers of modern SSMs, such as Mamba, quantifying theoretically the drastic improvement in performance from the previous generation of models, such as S4. Our theory not only motivates the success of modern selective state-space models, but also provides a solid framework to understand the expressive power of future SSM variants. In particular, it suggests cross-channel interactions could play a vital role in future improvements.

## 1   Introduction

Sequence-to-sequence blocks are fundamental components of modern deep learning models for language, images, video, audio, time series, and genomics. For the last five years,attention [Vaswani et al., 2017, Dosovitskiy et al., 2020] has been the dominant mechanism powering these architectures. However, competitive results have recently been achieved without attention, by using state-space models (SSMs): GPU-efficient linear recurrent sequence-to-sequence blocks stemming from S4 [Gu et al., 2021]. SSMs achieve state-of-the-art results on long-range-reasoning benchmarks [Tay et al., 2020] and show outstanding performance in various domain including vision [Nguyen et al., 2022], audio [Goel et al., 2022], biological signals [Gu et al., 2021], reinforcement learning [Lu et al., 2023] and online learning [Zucchet et al., 2023]. SSMs recently have gained significant interest in the community since their computational complexity scales linearly in sequence length, while attention scales quadratically; moreover, unlike other recurrent mechanisms such as LSTMs [Hochreiter and Schmidhuber, 1997] and GRUs [Cho et al., 2014], they can be efficiently parallelized on GPUs during training using parallel scans [Martin and Cundy, 2017, Smith et al., 2023].

38th Conference on Neural Information Processing Systems (NeurIPS 2024).

While standard SSMs were shown to be particularly powerful on signal processing tasks, their computation power is limited: the core sequential mechanism of S4 is equivalent to a convolution (filtering) [Li et al., 2022a]. This represents a drawback in challenging domains such as text and genetics, where the ability to select data efficiently in an input-dependent manner – i.e., perform content-based reasoning – is crucial (see [Wang et al., 2022, Fu et al., 2022, Arora et al., 2023]). Towards reaching this goal with recurrent models, various adaptations of S4 have been proposed in the last few months. Notably, Mamba [Gu and Dao, 2023] implements simple and efficient gating mechanisms on the S4 recurrence, unlocking input selectivity in the memory update. Mamba achieved state-of-the-art performance in various language modeling tasks while greatly improving the inference throughput. Similar ideas can be found in recent developments inspired by attention, such as RWKV [Peng et al., 2023], RetNet [Sun et al., 2023], Gateloop [Katsch, 2023], Gated Linear Attention (GLA) [Yang et al., 2023], and HGRN2 [Qin et al., 2024]. Very recently, De et al. [2024] surpassed the performance of Mamba with a gated RNN architecture – Griffin – based on an improved version of the LRU [Orvieto et al., 2023a], and [Feng et al., 2024] introduced minimal versions of GRU and LSTM as gated SSMs.

**Contributions**   At the core of the models discussed above is a time-varying dynamical system, where reasoning is performed through an efficient and parallelizable update *linear* in the hidden state. In this paper, we generalize the structure of such models, drawing a direct link to *controlled differential equations (CDEs)* [Young, 1936, Lyons, 1994, Kidger et al., 2020, Morrill et al., 2021, Fermanian et al., 2021, Salvi et al., 2022, Hoglund et al., 2023, Walker et al., 2024] and use tools from *rough path theory* [Lyons et al., 2007] to study expressivity.

1. In Sec. 3.1 we provide a framework for the analysis of (input-controlled) linear (in the hidden state) recurrences such as S4 and Mamba. This framework allows the use of powerful tools and results in the Rough Path Theory literature by casting a large family of SSMs as Linear CDEs driven by the two possibly nonlinear embeddings $X \mapsto \omega^X$ and $X \mapsto \xi^X$, defining gates. Appendices A and E provide a largely self-contained exposition of the key theoretical tools now available to us.

2. In Sec. 4 we fully characterize the closure (i.e. the class of functions which can be arbitrarily well approximated) of our generalized models. This provides a generalization of the results by Li et al. [2022b], Orvieto et al. [2023b], Wang and Xue [2023], who only consider the case of S4. The Mamba setting is more rich, complex, and relevant given the rising interest in selective SSMs.

3. We show (Thm. 4.2) that full expressivity can be obtained by training only a linear layer on a Linear CDE with random parameters, providing a direct link to kernel methods and reservoirs.

4. We point out (Thm. 4.3) that if the recurrence is diagonal, as the case for Mamba, the closure is strictly smaller than in the general dense case. Interestingly though, the closure is a peculiar set of filters that unlock some specific context-dependent processing. Full expressive power is recovered by stacking multiple SSMs without MLPs in between (Prop. 4.5).

Our framework not only provides significant theoretical insight regarding some recently proposed SSM architectures, but we also envision it to be a useful tool in analysing, and perhaps developing, future architectural advances.

## 2   State-space Models

We describe here the structure of the main SSMs-based strategies for processing length-$L$ input sequences of $d$ dimensional tokens: $x \in \mathbb{R}^{d \times L}$. We denote by $x_\ell$ the $\ell$-th column of $x$ (the $\ell$-th token) and by $x^i$ the $i$-th row of $x$ (time series for the $i$-th channel). We will write $A \cdot v$ for matrix-vector multiplication when this enhances comprehension, and use bold letters for "tensors" of order greater than 2 (such as $\mathbf{z} \in \times_i \mathbb{R}^{N_i \times L}$ introduced below).

### 2.1   Review of Modern SSMs

We start with a quick simplified recap of S4 [Gu et al., 2021], the first SSM proposed in the literature, and then describe recent improved variants such as Mamba (in particular, the S6 block) [Gu and Dao, 2023]. We restrict our focus to the recurrent mechanism and invite the reader to refer to the original papers for a description of the token-wise operations following and preceding each block.

**SSM basics and S4.** Most[1] SSMs [Gu et al., 2021, 2022] operate independently on input channels. Each time series $x^i \in \mathbb{R}^L$ is seen as the result of sampling a latent continuous-time signal $X^i : [0, 1] \to \mathbb{R}$ at multiples of a channel-dependent stepsize $\Delta_i > 0$: $X_{\Delta_i \ell}^i := X^i(\Delta_i \ell) = x_\ell^i$. In S4, each path $X^i$ produces a complex-valued hidden state signal $\mathbf{Z}_i : [0, 1] \to \mathbb{C}^{N_i}$ as

$$d\mathbf{Z}_{i;t} = A_i \cdot \mathbf{Z}_{i;t} \, dt + B \, X_t^i dt, \tag{1}$$

where $A_i = \text{diag}(a_{i,1}, a_{i,2}, \ldots a_{i,N})$ is channel-specific *diagonal* $N_i \times N_i$ complex valued matrix and $B \in \mathbb{C}^{N_i}$ is an input projection shared across input components $i \in [d]$. SSMs are based on a stable discretization of the continuous system above: each input sequence channel $x^i \in \mathbb{R}^L$ produces a sequence of hidden states $\mathbf{z}_i = [\mathbf{z}_{i;1} | \mathbf{z}_{i;2} | \ldots | \mathbf{z}_{i;L}] \in \mathbb{C}^{N_i \times L}$ as follows:

$$\mathbf{z}_{i;\ell} = \bar{A}_i \cdot \mathbf{z}_{i;\ell-1} + \bar{B}_i x_\ell^i, \tag{2}$$

where $\bar{A}_i$ and $\bar{B}_i$ are determined by the discretization technique and the channel-dependent stepsize $\Delta_i$. Under the commonly used Zero-Order Hold discretization[2],

$$\bar{A}_i = \exp(\Delta_i A_i), \quad \bar{B}_i = (\Delta_i A_i)^{-1}(\exp(\Delta_i A_i) - I)\Delta_i B \approx \Delta_i B. \tag{3}$$

Note from (2) that SSMs at inference time are equivalent to linear recurrent neural networks (RNNs). Yet, learning with gradient descent is performed on the continuous-time variables, unlocking stable signal propagation and alleviating vanishing gradients [Orvieto et al., 2023a, Zucchet and Orvieto, 2024]. Finally, at each channel $i$, the sequence of hidden states is mapped back to real numbers, and linear projections $C_i : \mathbb{C}^{N_i} \to \mathbb{R}$ are performed to produce an output a sequence of tokens $y \in \mathbb{R}^{d \times L}$ with the same dimensions as $x$:

$$y_\ell^i := C_i \cdot \mathbf{z}_{i;\ell}$$

To conclude, we point out that the transition matrices $A_i$ are often structured, i.e. initialized deterministically through HiPPO theory [Gu et al., 2020] in diagonal form. Common choices [Gu et al., 2022] are $a_{\cdot,n} = -\frac{1}{2} + i\pi n$ (S4D-Lin[3]) and $a_{\cdot,n} = -\frac{1}{2}$ (S4D-Real).

**Mamba.** As done in practice, let us consider all channels' hidden dimensions $N_i$ equal to $N$. The Selective SSM (S6) powering the Mamba architecture [Gu and Dao, 2023] augments S4 with input-controlled matrices:

$$\mathbf{z}_{i;\ell} = \bar{A}_i(x_\ell) \cdot \mathbf{z}_{i;\ell-1} + \bar{B}_i(x_\ell) x_\ell^i, \tag{4}$$

where the most crucial component (see in-context learning argument by Gu and Dao [2023]) is the dependency of the *diagonal* matrix $\bar{A}_i(x_\ell) \in \mathbb{R}^{N \times N}$ at timestamp $\ell$ on *all* input channels at timestamp $\ell$. This makes the operation $\bar{A}_i(x_\ell) \cdot \mathbf{z}_{i;\ell-1}$ effectively a *gate*. The dependency of $\bar{A}_i : \mathbb{R}^d \to \mathbb{R}^{N \times N}$ on the input is achieved efficiently by letting $\Delta_i$ in (3) be computed, at step $\ell$, as $\Delta_i(x_\ell)$ where

$$\Delta_i : \mathbb{R}^d \to \mathbb{R}, \quad \Delta_i(x) = \text{softplus}(\alpha_i \cdot x + \beta_i) \in \mathbb{R}$$

where $\cdot$ is the scalar product and [4] $\alpha_i \in \mathbb{R}^d$, $\beta_i \in \mathbb{R}$. Further, $\bar{B}_i : \mathbb{R}^d \to \mathbb{R}^N$ is computed via a, shared between channels, linear map $B \in \mathbb{R}^{N \times d}$ via $\bar{B}_i(x_l) = (B \cdot x_\ell) \Delta_i(x_\ell) \in \mathbb{R}^N$. Finally, each $\mathbf{z}_{i;\ell} \in \mathbb{R}^N$ is projected to $y_\ell^i \in \mathbb{R}$ via a matrix $C_i$. This step can also be done by means of output gating ($C_i$ function of the input), but we avoid this complication here as it can be seen as an architectural component outside the recurrence.

*Remark* 2.1. While each channel evolves separately, the laws of evolution are pointwise determined by all input features: $A_i$ and $B_i$ can be functions of $x_\ell$, and not just of $x_\ell^i$. We will discuss this in Sec. 4.3 after presenting our general results.

The RG-LRU [De et al., 2024] works similarly, yet processing all input channels at once with a diagonal recurrence. Gateloop [Katsch, 2023], GLA [Yang et al., 2023], and HGRN2 [Qin et al., 2024] leverage similar ideas, though they differ in parametrization and gating strategies.

---

[1]The LRU and S5 instead build a single recurrence operating on multidimensional (# channels) inputs.

[2]This corresponds to: (i) considering the continuous underlying signal $X_t$ to be piecewise constant, (ii) solving exactly ODE (1) and finally (iii) sampling $Z_t$ at the sample times of $X_t$. Refer to Appendix F.

[3]i denotes the imaginary unit $\sqrt{-1}$, not to be confused with the index $i$.

[4]In practice, Gu and Dao [2023]use a low-rank projection to construct the $\Delta_i$s. In other words, the matrix $[\alpha_i^\top, \alpha_2^\top, \ldots, \alpha_d^\top]$ is low rank. Another distinction that Mamba has compared to S4, is that the $A_i$ matrices controlling the recurrence on channels $i = 1, 2, \ldots d$, is now shared and can be thus referred to as $A$.

## 2.2 Known properties of (non-linear) recurrences

The expressiveness of standard nonlinear RNNs of the form $z_\ell = A\sigma(z_{\ell-1}) + Bx_\ell$, where $\sigma$ is a nonlinearity, has been extensively studied since the seminal work of Siegelmann and Sontag [1992], with recent contributions such as Korsky and Berwick [2019] and Hanson and Raginsky [2020]. In particular, Hanson and Raginsky [2020] proved that wide enough non-linear RNNs can approximate up to vanishing precision non-linear time-homogeneous systems of differential equations driven by input paths. The argument used here is based on the celebrated Barron's theorem [Barron, 1993] for approximation of continuous functions with neural networks with one hidden layer. Indeed, note that non-linear RNNs are recurrent perceptrons with one hidden layer, acting both on the state and the input [Tallec and Ollivier, 2018]. Instead, (selective) SSMs such as S4 and Mamba have transition map which is linear in the state – unlocking parallelization [Smith et al., 2023, Gu and Dao, 2023]. In the context of linear RNNs and non-selective SSMs, many results (classic and new) exist that characterize expressivity. Li et al. [2022b] showed that linear RNNs (i.e. S4-like recurrences) can approximate arbitrary convolution filters in the width limit. Further, Hanson and Raginsky [2019] proved that stacking exponentially (in the sequence length) many temporal convolution filters, chained together with ReLU activations, leads to approximation of arbitrary non-linear filters. Recent works [Orvieto et al., 2023b, Wang and Xue, 2023] prove the universality of linear recurrences (one layer) when equipped with a fixed (timestamp independent) point-wise MLP acting across the recurrence output, with intriguing connections to Volterra series [Boyd and Chua, 1985].

Mamba (alongside with gated linear attention variants e.g. Yang et al. [2023]) falls neither in the linear RNN nor the nonlinear RNN setting: its recurrence is linear on the hidden state (can be parallelized) but unlike S4, it is not linear time-invariant as the input controls the recurrence eigenvalues. In this paper, we are interested in this hybrid setting. It is worth noting that some work exploring Mamba's expressiveness has already been performed to study some interesting toy tasks [Jelassi et al., 2024] and to understand its limitation using the framework of formal language theory [Merrill et al., 2024]. Compared to these works, which outline interesting failure cases, this paper studies a more general class of models, allowing to identify how architectural choices impact expressivity.

## 3 SSMs as Linear CDEs

The crucial component that unlocks in-context learning and selectivity in modern SSMs is *the input-dependent state-to-state transition matrix [Gu and Dao, 2023], gating the hidden state* and thus allowing the system to filter out unnecessary context and remember relevant information indefinitely.

At the core of most modern SSMs is a a recurrence which is *linear in the hidden state, but potentially non-linear in the input*. This class includes many recent SSM-based or inspired models. Crucially, it does not contain classical RNNs, LSTMs, and GRUs – for which results are known and rely on the non-linear dependence of the hidden state in the update rule (Sec. 2.2). As we will shortly see, structure and features of (selective) SSMs can be studied within a unified, convenient, continuous-time framework (Linear Controlled Differential Equations). This allows to answer the following question:

> *"What is the most one can achieve using a recurrence which is*
> *linear in the hidden state and potentially non-linear in the input?"*

### 3.1 Linear CDEs

According to their continuous-time formulation, SSMs process input data sampled from a continuous path $X : [0,1] \to \mathbb{R}^d$, where $d$ is the number of channels. By $X_t^i$ we denote the input channel $i$, evaluated at time $t \in [0,1]$. More formally, we consider input trajectories in the separable Banach space $\mathbb{X} = C_{1,0}([0,1];\mathbb{R}^d)$ of absolutely continuous $\mathbb{R}^d$ dimensional paths. In this space, one can write $X = \int_0^t \dot{X}_s \, ds$ since $X \in L^1([0,1];\mathbb{R}^d)$; and the norm is $\|X\|_{1;[0,1]} := \int_0^1 |\dot{X}_s| \, ds$.

To model gates (see Mamba in (4)), we introduce two maps transforming the path $X$, which output trajectories $\omega^{\mathrm{X}}, \xi^{\mathrm{X}}$ living in potentially higher dimensions: $d_\omega$ and $d_\xi$

$$\omega : \mathbb{X} \to C_{1,0}([0,1];\mathbb{R}^{d_\omega}), \quad \xi : \mathbb{X} \to C_{1,0}([0,1];\mathbb{R}^{d_\xi})$$

where we used the shorthand notation $\omega^{\mathrm{X}} := \omega(X), \xi^{\mathrm{X}} = \xi(X)$. Akin the notation for the $i$-the channel of $X$ (i.e. $X^i$), we denote by $\omega^{\mathrm{X},i}$ the $i$-th channel in $\omega^{\mathrm{X}}$.

The role of the $\xi, \omega$ functions – which we denote *gating functions* – will be clear very soon.

**Definition 3.1** (Linear CDE). Fix $N \in \mathbb{N}$ (hidden-state dimension), matrices $A_1, ..., A_{d_\omega} \in \mathbb{R}^{N \times N}$ and $B \in \mathbb{R}^{N \times d_\xi}$. The hidden state $Z^{\mathrm{X}} := Z(X)$ is computed by the Linear CDE through a linear (in the hidden state) differential equation driven by $\omega, \xi$ – functions of the input path:

$$dZ_t^{\mathrm{X}} = \sum_{i=1}^{d_\omega} A_i Z_t^{\mathrm{X}} d\omega_t^{\mathrm{X},i} + Bd\xi_t^{\mathrm{X}}, \quad Z_0^{\mathrm{X}} = Z_0 \in \mathbb{R}^N \tag{5}$$

We show that both S4 and Mamba can be written in continuous-time as systems of parallel Linear CDEs. The key ingredient setting the two models apart is the choice of drivers $\omega$ and $\xi$. As in the preceding section, we will use the bold tensor notation to stress the parallel nature of these CDEs.

- **S4 is a Linear CDE:** It is sufficient to consider the setting $d = 1$, $N = N_i$. The S4 model [Gu et al., 2021] in this case is $dZ_t = A \cdot Z_t \, dt + B \, (X_t dt)$, where $A \in \mathbb{R}^{N \times N}$ is diagonal. This can be written as a Linear CDE with

$$\omega_t^{\mathrm{X}} = t \in \mathbb{R}, \qquad \xi_t^{\mathrm{X}} = \int_0^t X_s ds \in \mathbb{R}, \tag{6}$$

  since $d\omega_t^{\mathrm{X}} = dt$ and $d\xi_t^{\mathrm{X}} = X_t dt$. As the reader can promptly notice, here $\omega^{\mathrm{X}}$ is not a function of $X$ – this will have a crucial role in expressivity (see Sec. 4.2).

- **Mamba is a Linear CDE:** Recall from (4) that the recurrence inside Mamba (i.e. S6), can be written as $\mathbf{z}_{i;\ell} = \bar{A}_i(x_\ell) \cdot \mathbf{z}_{i;\ell-1} + \bar{B}_i(x_\ell)x_\ell^i$ where for generic timestamp features $x \in \mathbb{R}^d$, we have $\bar{A}_i(x) = \exp(\Delta_i(x)A_i)$, $\bar{B}_i(x) \approx (B \cdot x)\Delta_i(x)$ and $\Delta_i(x) = \mathrm{softplus}(\alpha_i \cdot x + \beta_i)$. Let us introduce a parameter $\delta > 0$, and consider $\tilde{\alpha}_i = \alpha_i/\delta$, $\tilde{\beta}_i = \beta_i/\delta$. Let us further approximate the softplus function with a ReLU ($\sigma(x) = \mathrm{ReLU}(x) \simeq \log(1 + e^x)$) to obtain $\Delta_i(x) = \sigma(\delta\tilde{\alpha}_i \cdot x + \delta\tilde{\beta}_i) = \sigma(\tilde{\alpha}_i \cdot x + \tilde{\beta}_i)\delta$. Therefore, as $\delta \to 0$, one has $\bar{A}_i(x) = \exp(\Delta_i(x)A_i) = \exp(\sigma(\tilde{\alpha}_i \cdot x + \tilde{\beta}_i)\delta A_i) \overset{\delta \to 0}{\to} 1 + \sigma(\tilde{\alpha}_i \cdot x + \tilde{\beta}_i)\delta A_i$, leading to the recurrence

$$\mathbf{z}_{i;\ell} = \mathbf{z}_{i;\ell-1} + A_i \cdot \mathbf{z}_{i;\ell-1} \, \sigma(\tilde{\alpha}_i \cdot x_\ell + \tilde{\beta}_i)\delta + (B \cdot x_\ell) \, x_\ell^i \sigma(\tilde{\alpha}_i \cdot x_\ell + \tilde{\beta}_i)\delta.$$

As we show formally in Appendix F, $\delta$ plays the role of the differential $dt$. The equation above is the Euler discretization of the differential equation

$$\mathbb{R}^N \ni d\mathbf{Z}_{i;t}^{\mathrm{X}} = A_i \cdot \mathbf{Z}_{i;t}^{\mathrm{X}} \, \sigma(\tilde{\alpha}_i \cdot X_t + \tilde{\beta}_i)dt + (B \cdot X_t) \, X_t^i \sigma(\tilde{\alpha}_i \cdot X_t + \tilde{\beta}_i)dt$$

where for each $i$, $\mathbf{Z}_i^{\mathrm{X}} : [0, 1] \to \mathbb{R}^N$. These are Linear CDEs with carefully chosen $\omega$ and $\xi$:

$$\boldsymbol{\omega}_{i;t}^{\mathrm{X}} = \int_0^t \sigma(\tilde{\alpha}_i \cdot X_s + \tilde{\beta}_i)ds \in \mathbb{R}, \quad \boldsymbol{\xi}_{i;t}^{\mathrm{X}} = \int_0^t X_s \, X_s^i \sigma(\tilde{\alpha}_i \cdot X_s + \tilde{\beta}_i)ds \in \mathbb{R}^d.$$

Note that here the $\boldsymbol{\xi}_i$ depend on higher powers of $X$'s dimensions, an intriguing feature connected to input gating [Hochreiter and Schmidhuber, 1997, Cho et al., 2014].

*Remark* 3.2 (Are hidden state components recurrently mixed?). In the Linear CDE defined by $dZ_t^{\mathrm{X}} = [\sum_{j=1}^{d_\omega} A_j d\omega_t^{\mathrm{X},j}] \cdot Z_t^{\mathrm{X}} + B \cdot d\xi_t^{\mathrm{X}}$ channels of the transformed input path $\omega^{\mathrm{X}}$ are mixed in a linear way in the recurrent step and stored in a shared hidden state $Z^{\mathrm{X}}$. This allows our framework to be more general compared to S4 and Mamba, where each channel of the input is processed individually, and *hidden states are later combined*. We know already from Merrill et al. [2024] that this distinction between hidden state mixing strategies is crucial for expressivity. Our discussion in Sec. 4.3 provides an in-depth look at the effects of separate channel processing, achieved in our framework by choosing the $A_i$s to be diagonal and non-zero only around a channel-specific portion.

## 4 Expressivity of Linear CDEs

Having established the connection between SSMs and Linear CDEs, we now provide an explicit characterization of the *uniform closure* of Linear CDEs, i.e. a description of all the functions from compact subsets of $\mathbb{X}$ to $\mathbb{R}$ that can be uniformly approximated at an arbitrary precision by a Linear CDE of the form given in (5).

## 4.1 Characterization of the closure

The proof of the main theorem we present here (Thm. 4.1) is involved and requires tools from Rough Path theory; we provide a full derivation in Appendix B with tools reviewed in the self contained Appendix E, as well as an introduction to the Signature approach in Section 4.5 and Appendix A. While familiarity with these concepts is not necessary to grasp the main results as stated, it is essential for a thorough comprehension.

In this subsection, Linear CDEs are analyzed in full generality – i.e., in the dense setting. While for efficiency reasons non-diagonal recurrences are rarely used in SSMs, our results allow to precisely characterize the Linear CDE hypothesis class (i.e. the *closure*), thus to the answer the question *"What is the most we can achieve from recurrences which are linear in the hidden state?"*. We show that Linear CDEs can model, at fixed time $t$, *arbitrary continuous functions* of the *whole* seen inputs.
Of course, understanding the diagonal setting with separate channel processing is of utmost importance in the current research landscape. The tools and results in this subsection allow us to directly discuss this case and compare it to the dense setting – this is presented in Sec. 4.3.

In this section, for any path $\gamma \in C_{1,0}([0,1],\mathbb{R}^{d_\gamma})$ and any sub-interval $[s,t] \subset [0,1]$, we denote by $\gamma_{[s,t]} \in C_{1,0}([0,1],\mathbb{R}^{d_\gamma})$ the path $\gamma_{[s,t]}(u) = \gamma_{t \wedge u} - \gamma_{s \wedge u}$, where $a \wedge b = \min(a,b)$.

**Theorem 4.1.** *Let $\mathbb{X} \subset C_{1,0}([0,1],\mathbb{R}^d)$ be compact and choose continuous gating functions $\omega, \xi$ such that[5] $\omega_t^{X,1} = t$ and $\omega_t^{X,2} = t^2$. Consider the Linear CDE model (5). Let $\Psi, \Phi$ be generic continuous functions from paths to real vectors:*

$$\Psi : C_{1,0}([0,1],\mathbb{R}^{d_\omega}) \to \mathbb{R}, \quad \Phi : C_{1,0}([0,1],\mathbb{R}^{d_\omega}) \to \mathbb{R}^{d_\xi}.$$

*There exist **dense** matrices $A_1, ..., A_{d_\omega}, B$ such that, after a fixed final linear projection $C \in \mathbb{R}^{1 \times N}$, the output $Y_t^X = CZ_t^X$ is arbitrarily close, uniformly on $\mathbb{X} \times [0,1]$, to*

$$\Psi(\omega_{[0,t]}^X) + \int_0^t \Phi(\omega_{[s,t]}^X) \cdot d\xi_s^X \tag{7}$$

*where $\cdot$ is the scalar product. Moreover $Y_t^X = CZ_t^X$ is itself of form (7).*

In Theorem 4.1, the $A_i$ are *dense* matrices constructed *ad hoc* for the proof. We show next that, with high probability, *random* Glorot-initialized matrices [LeCun et al., 2012] provide enough expressivity – one only has to choose the appropriate matrix $C$. This is a similar mechanism to the paradigm advocated in reservoir computing [Lukoveviius and Jaeger, 2009, Cuchiero et al., 2021a, Compagnoni et al., 2023]. The next result, however, is novel and of independent interest in Rough Path Theory.

**Theorem 4.2.** *Under the hypothesis of Theorem 4.1, pick $[A_j]_{n,n'} \overset{iid}{\sim} \mathcal{N}(0, \frac{1}{N})$ and $[Z_0]_n, [B]_{n,j} \overset{iid}{\sim} \mathcal{N}(0,1)$. For any functional $F : \mathbb{X} \times [0,1] \to \mathbb{R}$ of the form (7) and $\epsilon > 0$ it holds that*

$$\lim_{N \to \infty} \mathbb{P}\Big[\Big\{\exists C \in \mathbb{R}^{1 \times N} \text{ such that } \sup_{(X,t) \in \mathbb{X} \times [0,1]} |F(X,t) - CZ_t^X| \leq \epsilon\Big\}\Big] = 1.$$

## 4.2 Intuition on the closure result: role of gates

Roughly speaking, our result shows that dense Linear CDEs have drastically superior expressive power compared to dense linear RNNs. This contrast is to be attributed completely to the gate $\omega$.

**Warmup – Linear RNNs.** Thm 4.1 can be seen as a generalization of the *Universal Approximation for Linear RNNs* presented by Li et al. [2022b][Thm. 7] for generic gates $\omega, \xi$. In fact, their setting is restricted to $\omega_t^X = t$ and $\xi_t^X = \int_0^t X_s ds$. This is also the case for S4, S5 and the LRU – which are linear RNNs at test time. Then the only information contained in $\omega_{[s,t]}$ is the increment $t - s$ so that family (7) reduces to $\left\{(X,t) \mapsto \psi(t) + \int_0^t \phi(t-s) \cdot X_s ds\right\}$. This is, fundamentally, the set of linear filters on the input. As shown in Gu and Dao [2023], such processing is unable to adapt information flow in-context.

---

[5]These assumptions are of technical nature and can be relaxed at the cost of talking about *tree-like* equivalence of paths *cf.* Hambly and Lyons [2010]. Intuitively $\omega_t^{X,1} = t$ ensures general $\mathbb{X}$-uniformity while $\omega_t^{X,2} = t^2$ ensures $[0,1]$-uniformity.

**How does a Linear CDE process inputs?** Take without loss in generality[6] $\omega_t^X = X_t$. The first term in the function class $\{\Psi(X_{[0,t]}) + \int_0^t \Phi(X_{[s,t]}) \cdot d\xi_s^X\}$ is already enough to establish that the output $CZ_t^X$ is a nonlinear function of all previously seen inputs $X_{[0,t]} - \xi$ could be set to zero. The term $\Psi(X_{[0,t]})$ is however non-trivial only in the $Z_0 \neq 0$ case (*cf.* Appendix B): picking $\xi_t^X = 0 \in \mathbb{R}^N$ for all $t$ indeed leads to $dZ_t^X = \left[\sum_{i=1}^{d_\omega} A_i d\omega_t^{X,i}\right] Z_t^X$, which is evolving only if $Z_0 \neq 0$. While this setting is interesting and suggests a clear direction for future research, a case more similar to Mamba (see Sec. 4.3) is $Z_0 = 0$ and $\xi_t^X = \int_0^t X_s ds$. Here, we can approximate arbitrarily well outputs of the form

$$\left\{(X, t) \mapsto \int_0^t \Phi(X_{[s,t]}) \cdot X_s ds\right\} \tag{8}$$

where $\Phi$ is any continuous function of the input path, restricted to the portion $[s,t]$. This clearly shows that dense Linear CDEs are capable of *context-dependent filtering*: the output is again a linear combination of previously seen inputs, but weights are not predetermined as in Linear RNNs (S4) – they are a function of the context. A similar yet less powerful processing is happening in the diagonal setting, which we explore next.

## 4.3 The Diagonal Case

The choice of diagonal weights considerably restricts the family of learnable functionals. Intuitively the diagonal choice corresponds to running $N$ independent 1-dimensional systems, this absence of mixing between the different hidden dimensions is the main culprit for the loss of expressivity. The full power can however be recovered by *chaining* the diagonal schemes (*cf.* Prop. 4.5).

**Theorem 4.3** (Diagonal Case). *If the matrices $A_1, ..., A_{d_\omega}$ are constrained to be diagonal, the requirements $\omega_t^{X,1} = t$, $\omega_t^{X,2} = t^2$ can be dropped and the closure reduces to*

$$\left\{(X, t) \mapsto \psi(\omega_t^X) + \int_0^t \phi(\omega_t^X - \omega_s^X) \cdot d\xi_s^X\right\} \tag{9}$$

*for continuous $\psi : \mathbb{R}^{d_\omega} \to \mathbb{R}$ and $\phi : \mathbb{R}^{d_\omega} \to \mathbb{R}^{d_\xi}$.*

Compared to the dense setting, the effect of diagonality is pretty clear. Let us again assume $\omega_t^X = X_t$ and $\xi_t^X = \int_0^t X_s \, ds$ for simplicity. While $\int_0^t \Phi(X_{[s,t]}) \cdot X_s \, ds$ unlocks filtering based on the entire trajectory $X_{[s,t]}$, the diagonal case term $\int_0^t \phi(X_t - X_s) \cdot X_s \, ds$ indicates that filtering coefficients can only be chosen by comparing two elements of the (potentially transformed through $\omega$) input sequence. While this precise and tight result reveals a pitfall of diagonal recurrence, it also brings about an interesting connection to attention [Vaswani et al., 2017], where only a finite number of tokens are compared at each layer. While a smart choice of gating functions $\omega, \xi$ can improve on the learned nonlinear filtering strategy (e.g. based on filtered input versions as in Mamba), our theory reveals the fundamental processing discrepancy compared to the dense setting, a property which was also explored in recent literature [Merrill et al., 2024] using different tools.

As already noted, on top of diagonality, recent SSMs also mix inputs as linear combinations of independently run channel-dependent systems. This slightly modifies the function class as:

**Corollary 4.4** (Mamba Case). *In the Mamba setting, the closure reduces to*

$$\left\{(X, t) \mapsto \sum_{i=1}^{d_\omega} \psi_i(\omega_t^{X,i}) + \sum_{i=1}^{d_\omega} \int_0^t \phi_i(\omega_t^{X,i} - \omega_s^{X,i}) \, d\xi_s^{X,i}\right\} \tag{10}$$

*for continuous $\psi_i : \mathbb{R} \to \mathbb{R}$ and $\phi_i : \mathbb{R} \to \mathbb{R}$.*

## 4.4 Chaining Diagonal CDEs

Fortunately it is possible to re-gain expressivity without sacrificing the computational advantages of diagonal schemes through *chaining*. This means driving a new Linear CDE by the solution of a previous Linear CDE, and repeating this procedure $K$ times (*cf.* Appendix C).

---

[6]Let us ignore the terms $t, t^2$ at this stage; their role is purely technical due to possible path equivalences.

**Proposition 4.5.** *Assume a compact $\mathbb{X} \subset C_{1,0}([0,1];\mathbb{R}^d)$. For any functional $F : \mathbb{X} \times [0,1] \to \mathbb{R}$ of the form (7) and $\epsilon > 0$ there exists a sequence of linear maps $W_k \in \mathbb{R}^{M_k \times N_k}$,* diagonal *weights $A_i^{(k)}$ and $B^{(k)}$ for the following family of* chained *diagonal Linear CDEs*

$$Z_t^{0,X} \equiv 0, \quad dZ_t^{k+1,X} = \sum_{i=1}^{d+M_k} A_i^{(k+1)} Z_t^{k+1,X} d \begin{bmatrix} W_k Z^{k,X} \\ X \end{bmatrix}_t^i + B^{(k+1)} dX_t \in \mathbb{R}^{N_{k+1}}, \quad (11)$$

*such that* eventually, *as $k \to \infty$, there exists $C_k \in \mathbb{R}^{1 \times N_k}$ with $\displaystyle\sup_{(X,t)\in\mathbb{X}\times[0,1]} |F(X,t) - C_k Z_t^{k,X}| \leq \epsilon$.*

Intuitively, with chaining one recovers the mixing between the input dimensions which was so important for the expressiveness of *dense* Linear CDEs. This does not happen immediately but crucially depends on the length of the chain, the result in fact tells us that the recovery holds for *long enough* chains (and big enough hidden states). In Appendix C.2.1 we argue that the same conclusions hold also in the Mamba setting with non-linear gates.

### 4.5 Proof Idea - Signature expansion

In this section, we introduce the primary tools, objects, and techniques from Rough Path theory used in our proofs. Given their technical nature, we have chosen to present the main results of this paper without explicit reference to them, allowing readers to understand the work without needing specialized knowledge. For those interested in the finer details, here we provide a brief overview and refer to Appendix A for additional details and references.

To study the expressivity of Linear CDEs it is convenient to introduce the so-called *signature transform* [Lyons et al., 2007, Kidger et al., 2019, Fermanian et al., 2023], a classical path-transform from stochastic analysis. The main reason for doing so is that, as a simple consequence of the Stone-Weirestrass theorem, linear functionals on the signature provide the essential building blocks (analogous to monomials on Euclidean spaces) to approximate continuous functions on path space.

Consider a path $\gamma \in C_{1,0}([0,1];\mathbb{R}^{d_\gamma})$ and define as $\mathbb{W}_{d_\gamma}$ the set of *words* (*i.e.* ordered sequences) in $\{1, \ldots, d_\gamma\}$[7]. The signature transform is the following infinite collection of scalar iterated integrals

$$\text{Sig}(\gamma)_{s,t} := \left( \text{Sig}(\gamma)_{s,t}^{(I)} \right)_{I \in \mathbb{W}_{d_\gamma}}, \quad \text{Sig}(\gamma)_{s,t}^{(I)} := \int_{s<u_1<\ldots<u_n<t} \dot{\gamma}_{u_1}^{(i_1)} \ldots \dot{\gamma}_{u_n}^{(i_n)} du_1 \ldots du_n.$$

A classical result from rough path theory states that a Linear CDE can be expanded explicitly as an (infinite) linear combination of terms in the signature of the driving path.

**Proposition 4.6.** *For any choice of matrices $A_1, \ldots, A_{d_\omega}$ and $B$, the unique solution Linear CDE (5) is, using the notation $A_I := A_{i_n} \ldots A_{i_1}$, given by*

$$Z_t^X = \sum_{I \in \mathbb{W}_{d_\omega}} A_I Z_0 \, Sig(\omega^X)_{0,t}^{(I)} + \sum_{i=1}^{d_\xi} \sum_{I \in \mathbb{W}_{d_\omega}} A_I B_i \int_0^t Sig(\omega^X)_{s,t}^{(I)} d\xi_s^{X,i} \in \mathbb{R}^N. \quad (12)$$

Notice that the previous result *does not* rely on any assumptions on the nature of $Z_0$, $A_i$, and $B$; for any such choice the result is a *time-independent linear map* on a *feature vector* $T(X)_{0,t} := (\text{Sig}(\omega^X)_{0,t}^{(I)}, \int_0^t \text{Sig}(\omega^X)_{s,t}^{(I)} d\xi_s^{X,i})_{(I,i)}$, where the index $(I,i)$ runs over $\mathbb{W}_{d_\omega} \times \{1, \ldots, d_\xi\}$.

The main takeaway is that any linear projection $Y_t^X := C Z_t^X$ is written as an (infinite) linear combination of the terms in $T(X)_{0,t}$. This means that the expressive power of such schemes is almost *completely determined* by the gate $\omega$, which is the only path of which high order information is taken into consideration through its full *signature*. It is evident then that the classical choice of input-independent $\omega$ (i.e. $\omega_t^X = t$) then *precludes* the use of higher order statistics of $X$.

## 5 Path-to-Path Learning

In Section 4, we showed that Linear CDEs (and chained Mamba) can model, *at fixed time $t$*, arbitrary continuous functions of the whole seen inputs. Assume wanting to learn a functional of type

---

[7]So when we write $I \in \mathbb{W}_{d_\gamma}$ we mean $I = i_1 i_2 \cdots i_M$ for some integer $M \geq 0$ and letters $i_m \in \{1, \ldots, d_\gamma\}$, then $Ii$ will be the word $i_1 i_2 \cdots i_M i$ for $i \in \{1, \ldots, d_\gamma\}$.

$(X, t) \mapsto \Psi_t(\omega^{\mathrm{X}}_{[0,t]})$ where the map $\Psi_t$ is changing with time as well, this is an example of a general continuous path-to-path model. Note in particular how this is a more general family than the one of Theorem 4.1, where the maps $\Psi$ and $\Phi$ are fixed once and for all. Here, we discuss how passing the hidden state through a multi-layer perceptron (MLP), instead of just a linear readout (matrix $C$), allows us to efficiently approximate this richer class, interpolating between the $\Psi_t$s.

As shown in Orvieto et al. [2023b], classical SSMs followed by an MLP are universal on sequences. Given their nature of input-independent convolutions, their construction defers all reasoning to the MLP acting on the output: in S4, the SSM is simply providing an input compression – with no added reasoning. In our setting, we instead characterized the processing power of input-controlled (dense or diagonal) SSMs precisely, showing how it greatly surpasses linear filtering. For this reason, *the computational burden for an MLP action on a general Linear CDE would be greatly diminished* and its actual function reduced to an interpolation of the maps $\Psi_t$. We defer the proof to Appendix D.

**Proposition 5.1.** *Fix a compact set $\mathbb{K} \subseteq \mathbb{X}$ and continuous $\omega, \xi$ with $\omega^{X,1}_t \equiv t$. Then for all $\epsilon > 0$ and all* causal [8] *continuous mapping $G : C_{1,0}([0,1], \mathbb{R}^{d_\omega}) \times [0,1] \to \mathbb{R}$ there exist an integer $N \geq 0$, some MLP $F : \mathbb{R}^N \to \mathbb{R}$, and parameters $Z_0 \in \mathbb{R}^N, A_i \in \mathbb{R}^{N \times N}, B \in \mathbb{R}^{N \times d_\xi}$ such that*

$$\sup_{(X,t) \in \mathbb{K} \times [0,1]} |F(Z^X_t) - G(\omega^X, t)| < \epsilon. \tag{13}$$

# 6 Empirical Validation

Code to reproduce all of our experiments can be found at:

https://github.com/Benjamin-Walker/selective-ssms-and-linear-cdes

**Datasets.** The first task is based on a dataset from Walker et al. [2024] where the aim is to predict terms in the anti-symmetric part of the input path's signature. The dataset's objective aligns with the proofs of Theorem 4.1 and 4.2, which characterise the closure using the path's signature. We created two datasets with dimensions 2 and 3 respectively. The increment in each channel at each step is an integer-rounded sample from a standard Normal distribution. The 2D dataset's target is an area integral $\int_0^1 \int_0^v dX^1_u dX^2_v$, and the 3D dataset's target is a volume integral $\int_0^1 \int_0^w \int_0^v dX^1_u dX^2_v dX^3_w$.

The second task is the $A_5$ benchmark from Merrill et al. [2024]. It tests models on state-tracking, a crucial ability for tasks involving permutation composition, such as chess. The dataset comprises sequences from the group of even permutations on five elements, $A_5$, where the target is the cumulative composition of all preceding permutations. Datasets vary by sequence length, ranging from 3 to 20.

**Models.** On the anti-symmetric signature task, we considered seven models: (i-ii) S4 or Mamba recurrence with linear readout, (iii-iv) two stacked S4 or Mamba recurrences with a linear mixing layer in-between and a linear readout, (v-vi) two stacked S4 or Mamba recurrences with a linear mixing layer + GeLU in-between, and a linear readout, (vii) a linear CDE with gates $\omega^X_t = \xi^X_t = (t, X_t)$ and a linear readout. All state space models have **trainable** matrices in their recurrences, whereas the linear CDE is using **fixed** matrices. All models use a hidden dimension of $256$, with the state space models using a state dimension of $256$. The state space models are trained using gradient descent with a batch size of 32 and Adam with a learning rate of $10^{-4}$. The output from the linear CDE's recurrence is obtained using the Tsit5 adaptive ODE solver, with an absolute and relative tolerance of $10^{-2}$. The linear CDEs linear readout is optimised via ordinary least squares.

On the $A_5$ benchmark, we consider five models: a linear CDE, a RNN, a transformer, and S4 and Mamba recurrences. All models use an embedding layer followed by a series of blocks that combine the sequence-to-sequence model, linear mixing with a non-linear activation function, layer normalization, and a residual connection. Furthermore, all models have **trainable** matrices in their recurrences and are trained using batch gradient descent with a batch size of 32 and AdamW with a weight decay of $0.01$. The RNN, transformer, S4, and Mamba use a hidden dimension of $1024$, with the state space models using a state dimension of $64$ and the transformer using $64$ heads. Due to memory constraints, the linear CDE uses a hidden dimension of $256$. Note that the IDS4 recurrence introduced by Merrill et al. [2024] corresponds directly to a linear CDE with dense transition matrices, hence we did not include the model as a baseline.

---

[8] A *causal* map is one which does not "look in the future" *cf.* Appendix D.

**Results.** Results on the anti-symmetric signature prediction task (Fig. 1) empirically demonstrate a number of the theoretical results presented in this paper. Firstly, as discussed in Sec. 4.2, recurrences which are linear in the input, such as S4, require a non-linearity in-between the layers to perform well. Furthermore, as stated in Thm. 4.3 and Prop. 4.5, even if the recurrence is non-linear in the input, such as Mamba, the expressivity of models with diagonal matrices is improved by stacking. Additionally, the inclusion of the non-linearity in-between the Mamba layers does not improve performance, as the recurrences themselves are expressive enough. Finally, as stated in Thm. 4.2, dense matrices can achieve strong expressivity with random initialization, no stacking, and only a trainable linear readout.

Fig. 2 is a plot of the results on the $A_5$ benchmark. The figure shows that the number of blocks S4, Mamba, and the transformer require to achieve greater than 90% validation accuracy grows with the sequence length. In fact, given our model architecture, none of these models could achieve greater than 90% validation accuracy for sequences of length twenty[9]. On the other hand, the RNN and Linear CDE are able to achieve greater than 90% validation accuracy for all lengths considered using only one block. These empirical results further validate Thm. 4.3 and Prop. 4.5: there exists a gap in expressivity between diagonal and dense transition matrices, and stacking is required to recover the expressivity. Furthermore, they provide empirical evidence that even for simple state-tracking problems, the number of blocks required can grow quickly with sequence length.

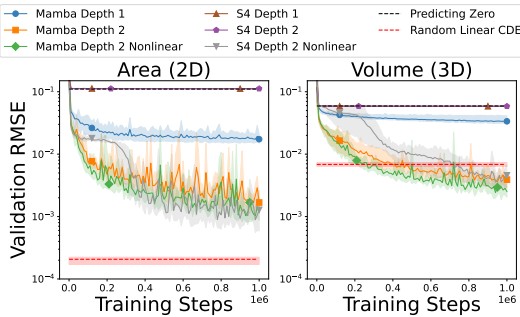

Figure 1: Comparison of the Linear CDE, Mamba, and S5 on the anti-symmetric signature prediction tasks. For each model, we plotted the mean and range of the validation accuracy over 5 independent runs.

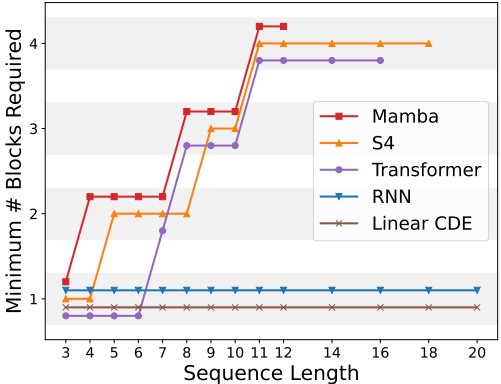

Figure 2: For each sequence length, the plot shows the minimum number of blocks required to achieve at least 90% validation accuracy, with each grey band corresponding to a number of blocks. Missing points mean the model did not achieve at least 90% validation accuracy with 4 blocks or less.

## 7 Conclusions

This paper explores Linear CDEs, a model family that extends both classical and modern SSM architectures, including recent gated RNNs. Using Rough Paths theory, we have characterized their uniform closure, generalizing the results of Li et al. [2022b] for linear RNNs. We precisely identified the advantages of input-controlled transition dynamics, which allow you to capture high-order statistics of the input as opposed to just the linear ones extracted by convolutions. While dense models reach full expressiveness, imposing diagonality (e.g. Mamba) weakens the model capabilities. This is in direct contrast to S4, where dense and diagonal settings share the same closure. Our analysis lays the theoretical foundation for analyzing the expressive power of future SSM variants and hints at non-diagonality as a potential source of improvements. We believe that light and efficient channel mixing in the recurrent block might already unlock the modeling of higher-order statistics.

**Limitations.** The current framework examines expressivity from a continuous-time perspective with real-valued inputs. The RNN literature suggests that finite precision often plays a significant role in practice, making it an interesting direction for future exploration. Additionally, although dense transition matrices are shown to be theoretically and empirically more expressive than diagonal ones, their increased computational cost makes them impractical for large-scale models.

---

[9]Merrill et al. [2024] show that with tailored architectures, S4, Mamba, and a Transformer can achieve over 90% validation accuracy on length-20 sequences using 4 blocks. However, in our experiments we wished to keep the architecture consistent between models, only changing the sequence-to-sequence model.

## Acknowledgments

Antonio Orvieto acknowledges the financial support of the Hector Foundation. Terry Lyons was funded in part by the EPSRC [EP/S026347/1], in part by The Alan Turing Institute [EP/N510129/1], in part by the Defence and Security Programme, in part by the Office for National Statistics, and in part by the Hong Kong Innovation and Technology Commission (InnoHK Project CIMDA). Benjamin Walker was funded by the Hong Kong Innovation and Technology Commission (InnoHK Project CIMDA). The authors would like to acknowledge the use of the University of Oxford Advanced Research Computing (ARC) facility in carrying out this work. http://dx.doi.org/10.5281/zenodo.22558

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

# A  Introduction to Signatures

This initial section of the Appendix is devoted to a brief introduction to the topic of Signature Transform. For a more in-depth account we refer the interested reader to Cass and Salvi [2024].

## A.1  Intuition - Controlled Differential Equations

In the simplest setting of smooth paths a CDE is a differential equation of form

$$\frac{dZ_t}{dt} = F\Big(\frac{dX_t}{dt}, Z_t\Big), \quad Z_0 \in \mathbb{R}^n$$

where $X : [0,1] \to \mathbb{R}^d$ is a known smooth path to which we refer as *control*, $Z_0$ the known initial condition and $Z : [0,1] \to \mathbb{R}^n$ the unknown solution.

The natural generalization is the following: assume to have two spaces $\mathbb{R}^{d_x}$ and $\mathbb{R}^{d_z}$, $X \in C_1([0,1];\mathbb{R}^{d_x})$, $Z \in C_1([0,1];\mathbb{R}^{d_z})$, $F : \mathbb{R}^{d_z} \to \mathcal{L}(\mathbb{R}^{d_x}, \mathbb{R}^{d_z})$ and $Z_0 \in \mathbb{R}^{d_z}$. We say that $(Z, X, F, Z_0)$ satisfy the CDE

$$dZ_t = F(Z_t)dX_t, \quad Z_0 \in \mathbb{R}^{d_z}$$

whenever

$$Z_t = Z_0 + \int_0^t F(Z_s)dX_s$$

The theory of *Rough Paths* has its origins in the study of such types of differential equations and provides a theoretical framework to define and work in rough settings i.e. when $X$ is not kust BV but even $\alpha$-Hölder for $\alpha \in (0,1)$ *cf.* Friz and Victoir [2010].

Assume thus to have the CDE

$$dZ_t = \sum_{i=1}^{d_x} V_i(Z_t)dX_t^i, \quad Z_0 \in \mathbb{R}^{d_z}$$

for sufficiently regular vector fields $V_i$ and $X \in C_1([0,1];\mathbb{R}^{d_x})$. Given a smooth $f : \mathbb{R}^n \to \mathbb{R}$, by the change of variable formula (*i.e.* fundamental theorem of calculus) we have

$$f(Z_t) = f(Z_0) + \sum_{i=1}^{d_x} \int_0^t V_i f(Z_s)dX_s^i$$

where $V_i f(z) := df_y[V_i(z)]$. Iterating this procedure on the $V_i f$s, *i.e.* substituting in the previous equation the analogously obtained equality

$$V_i f(Z_s) = V_i f(Z_0) + \sum_{j=1}^{d_x} \int_0^s V_j(V_i f)(Z_u)dX_u^j,$$

we get

$$f(Z_t) = f(Z_0) + \sum_{i=1}^{d} V_i f(Z_0) \int_0^t dX_s^i + \sum_{i,j=1}^{d} \int_0^t \int_0^s V_j V_i f(Z_u)dX_u^j dX_s^i$$

Keeping with this procedure for $N$ steps we get

$$f(Z_t) = f(Z_0) + \sum_{k=1}^{N} \sum_{|I|=k} V_I f(Z_0) \int \cdots \int_{s<u_1<\cdots<u_k<t} dX_{u_1}^{i_1} \cdots dX_{u_k}^{i_k} + R_N(t)$$

where $I = (i_1, \ldots, i_k)$ runs through the multi-indices, $V_I f := V_{i_1} V_{i_2} \ldots V_{i_k} f$ and

$$R_N(t) := \sum_{|J|=k+1} \int \cdots \int_{s<u_1<\cdots<u_{k+1}<t} V_J f(Z_{u_1})dX_{u_1}^{j_1} \cdots dX_{u_{k+1}}^{j_{k+1}}$$

As one can imagine, under reasonable regularity assumptions, the remainder goes to $0$ as $N \to \infty$ and at the limit

$$f(Z_t) = f(Z_0) + \sum_{k=1}^{\infty} \sum_{|I|=k} V_I f(Z_0) \int \cdots \int_{s < u_1 < \cdots < u_k < t} dX_{u_1}^{i_1} \cdots dX_{u_k}^{i_k}$$

This is a remarkable result: to know the solution $Z_t$ to the original CDE it suffices to know the quantities $V_I f$ for all multi-indices and $f$ in the coordinate maps, together with the iterated integrals

$$\mathrm{Sig}(X)_{s,t}^I := \int \cdots \int_{s < u_1 < \cdots < u_k < t} dX_{u_1}^{i_1} \cdots dX_{u_k}^{i_k}.$$

This observation is at the core of *Rough Path Analysis*, the theory can in a sense be considered an extreme development of it. The collection of iterated integrals, the *Signature*, will be the main for our analysis.

In Appendix E we expand and make rigorous the arguments of this section in the case of affine vector fields.

## A.2   Basic Definitions

Denote by $(\mathbb{R}^d)^{\otimes n} := \mathbb{R}^d \otimes \cdots \otimes \mathbb{R}^d$ the tensor product of $n$ copies $\mathbb{R}^d$, set $(\mathbb{R}^d)^{\otimes 0} := \mathbb{R}$. Let $T(\mathbb{R}^d) := \bigoplus_{k=0}^{\infty} (\mathbb{R}^d)^{\otimes k}$ be the tensor algebra equipped with sum and tensor product.

**Definition A.1.** Let $\{e_1, \ldots, e_d\}$ be the canonical basis of $\mathbb{R}^d$, then

$$\{e_{i_1} \otimes \cdots \otimes e_{i_k} : (i_1, \ldots, i_k) \in [d]^k\}$$

is a basis of $(\mathbb{R}^d)^{\otimes k}$. We equip $(\mathbb{R}^d)^{\otimes k}$ with the inner product $\langle \cdot, \cdot \rangle_{(\mathbb{R}^d)^{\otimes k}}$ defined on basis elements as

$$\langle e_{i_1} \otimes \cdots \otimes e_{i_k}, e_{j_1} \otimes \cdots \otimes e_{j_k} \rangle_{(\mathbb{R}^d)^{\otimes k}} = \delta_{(i_1, \ldots, i_k)}^{(j_1, \ldots, j_k)}$$

We extend this product to $T(\mathbb{R}^d)$ by

$$\langle A, B \rangle_{T(\mathbb{R}^d)} := \sum_{k=0}^{\infty} \langle a_k, b_k \rangle_{(\mathbb{R}^d)^{\otimes k}}$$

where $A = (a_0, a_1, \ldots)$ and $B = (b_0, b_1, \ldots)$

Note how for any $A, B \in T(\mathbb{R}^d)$ we have $\langle A \otimes e_i, B \otimes e_j \rangle_{T(\mathbb{R}^d)} = \langle A, B \rangle_{T(\mathbb{R}^d)} \langle e_i, e_j \rangle_{\mathbb{R}^d}$, we refer to this as to the *coproduct property*.

**Definition A.2** (Infinite Tensor Algebra)**.** The infinite tensor algebra is defined as the space $T((\mathbb{R}^d)) := \prod_{k=0}^{\infty} (\mathbb{R}^d)^{\otimes k}$ equipped with the operations $+$ and $\otimes$ which act in the natural algebraic way; its elements are called *tensor series*.

It is easily seen that $(T((\mathbb{R}^d)), +, \otimes)$ is an algebra with unit $\mathbf{1} = (1, 0, 0, \cdots)$ and we can endow it with a natural product which inherits the coproduct property.

Another point of view could be taken on the definitions of these spaces, one that we will prefer later on. If we define $\mathbb{W}_d$ to be the set of words in $d$ letters then $T((\mathbb{R}^d)) \sim \mathbb{R}^{\mathbb{W}_d}$, $T(\mathbb{R}^d)$ is the subset of such functions with finite support and

$$\langle A, B \rangle_{T(\mathbb{R}^d)} = \sum_{I \in \mathbb{W}_d} A_I B_I = \sum_{k=0}^{\infty} \sum_{|I|=k} A_I B_I$$

where $|I|$ is the length of the word $I$. The empty word, the only one with length $0$, is denoted by $()$ and corresponds to the basis element of $(\mathbb{R}^d)^{\otimes 0}$. In this view the tensor product coincides with concatenation of words accordingly distributed and the closure of $T(\mathbb{R}^d)$ with respect to its product is just the $l^2$ space $l^2(\mathbb{W}_d)$.

**Definition A.3** (Signature). Given $\gamma \in C_{1,0}([0,1];\mathbb{R}^d)$ and $s,t \in [0,1]$ s.t. $s \leq t$, the signature $Sig(\gamma)_{s,t} \in T((\mathbb{R}^d))$ of the path $\gamma$ over $[s,t]$ is defined as

$$\text{Sig}(\gamma)_{s,t} := (1, \int_{s<u_1<t} d\gamma_{u_1}, \cdots, \int_{s<u_1<\cdots<u_k<t}\cdots\int d\gamma_{u_1} \otimes \cdots \otimes d\gamma_{u_k}, \cdots)^{10} \tag{14}$$

Equivalently $\text{Sig}(\gamma)_{s,t}$ is that element of $l^2(\mathbb{W}_d)$ defined recursively on words as

$$\text{Sig}(\gamma)^{()}_{s,t} = 1, \quad \text{Sig}(\gamma)^{Ij}_{s,t} = \int_s^t \text{Sig}(\gamma)^I_{s,r} d\gamma^j_r. \tag{15}$$

## A.3 Notable Results

Here we present some notable results of which we will make use through the paper. We omit the proofs if they can be easily found in the suggested references.

The first result is about bounding the norm of Signature entries:

**Proposition A.4** (Factorial Decay Rate). *Given $\gamma \in C_{1,0}([0,1];\mathbb{R}^d)$, for all $k \geq 1$ and $s,t \in [0,1]$ s.t. $s \leq t$ one has*

$$\left\| \int_{s<u_1<\cdots<u_k<t}\cdots\int d\gamma_{u_1} \otimes \cdots \otimes d\gamma_{u_k} \right\|_{(\mathbb{R}^d)^{\otimes k}} \leq \frac{\|\gamma\|^k_{1-var,[s,t]}}{k!} \tag{16}$$

The most important fact about Signature is that it acts as the basis for a Taylor expansion in path space. In fact just as finite linear combinations of monomials are dense in the continuous functions with a compact input set, finite linear combinations of Signature entries are dense in continuous functions from compact path-spaces:

**Theorem A.5** (Universal Approximation Fermanian [2020]). *Fix $K \subset C_{1,0}([0,1];\mathbb{R}^{d+1})$ compact such that for any $\gamma \in K$ it holds $\gamma^1_t = t$. For any $F \in C^0(K;\mathbb{R})$ and $\epsilon > 0$ there is an integer $N \geq 0$ such that*

$$\sup_{\gamma \in K} |F(\gamma) - \sum_{|I| \leq N} \alpha_I Sig(\gamma)^I_{0,1}| \leq \epsilon \tag{17}$$

*for some finite sequence $(\alpha_I)_{|I| \leq N}$ of real numbers.*

*Remark* A.6. There is no magic in this result, it is just an application of Stone-Weiestrass enabled by the rich algebraic structure of iterated integrals, studied originally in Chen [1958].

We will need to restrict some paths to sub-intervals of $[0,1]$ in such a way to still be able to consider them meaningfully as elements of $C_{1,0}([0,1];\mathbb{R}^d)$, this is done in the following way:

**Definition A.7.** Given any path $\gamma \in C_{1,0}([0,1];\mathbb{R}^d)$ we define its restriction on a sub-interval $[s,t] \subseteq [0,1]$ as the path $\gamma_{[s,t]} \in C_{1,0}([0,1];\mathbb{R}^d)$ with values

$$\gamma_{[s,t]}(r) := \begin{cases} 0 & \text{if } r < s \\ \gamma_r - \gamma_s & \text{if } s \leq r \leq t \\ \gamma_t - \gamma_s & \text{if } r > t \end{cases} \tag{18}$$

This definition is such that the following important equation holds

$$\text{Sig}(\gamma)_{s,t} = \text{Sig}(\gamma_{[s,t]})_{0,1} \tag{19}$$

With the right augmentation of the paths one can see that the Signature distinguishes between different sections of paths, this will be crucial for some of the original results presented in this work.

**Lemma A.8.** *Assume $\omega, \gamma \in C_{1,0}([0,1];\mathbb{R}^{d+2})$ with $\omega^1_t = \gamma^1_t \equiv t$ and $\omega^2_t = \gamma^2_t \equiv t^2$. Then*

$$Sig(\omega)_{s,t} = Sig(\gamma)_{s',t'} \iff \omega_{[s,t]} = \gamma_{[s',t']} \tag{20}$$

---

[10]Here the integral is intended in the Riemann-Stjeltes sense.

*Proof.* The *if* part is follows from $\mathrm{Sig}(\gamma)_{s,t} = \mathrm{Sig}(\gamma_{[s,t]})_{0,1}$. For the *only if* part, If $s = s'$ and $t = t'$ the statement holds; this is because if the signatures over the time interval $[s, t]$ of two time-augmented paths are equal, then the two paths must be equal on $[s, t]$. We now show that augmenting the path with $t^2$ and imposing equality of signatures, implies $s = s'$ and $t = t'$, which will in turn allow us to conclude the proof by the previous remark. Assume $\mathrm{Sig}(\omega)_{s,t} = \mathrm{Sig}(\gamma)_{s',t'}$, in particular we must have

$$\int_s^t d(r^2) = t^2 - s^2 = (t')^2 - (s')^2 = \int_{s'}^{t'} d(r^2) \tag{21}$$

$$\int_s^t d(r) = t - s = t' - s' = \int_{s'}^{t'} d(r) \tag{22}$$

which reduces to the system

$$\begin{cases} t^2 - s^2 = (t')^2 - (s')^2 \\ t - s = t' - s' \end{cases} \quad \begin{cases} t + s = t' + s' \\ t - s = t' - s' \end{cases} \quad \begin{cases} 2t = 2t' \\ 2s = 2s' \end{cases}$$

Hence it must be true that $t = t'$ and $s = s'$. $\qquad\square$

## B  Expressivity

### B.1  Model Recap

In the body of the paper we have presented the main results with the simplified assumption of $Z_0^X = 0$ or at best $Z_0^X = Z_0$ *i.e.* with an initial value independent from the input. In this appendix we will carry on the proofs in a more general setting in which $Z_0^X$ is allowed to be input-dependent, as previously discussed the choice of initial value is, in contrast to the classical setting, meaningful inasmuch it allows to approximate linear maps on the signature of $\omega_{[0,1]}^X$. In order to do so we have to introduce a new gate, the initial value gate, in the form of a map

$$
\begin{aligned}
(\cdot)_0 : \mathbb{X} &\to \mathbb{R}^{d_0} \\
X &\mapsto X_0
\end{aligned}
\tag{23}
$$

Despite the notation, there is no reason why $(X)_0$ should be the initial value of the path $X$, one should think of this map as the one summarizing the data which still matters for the task but which does not have a time-series nature.

To recapitulate, the general setting of our models is the following: a topological *input space* space $\mathbb{X}$,

$$(\cdot)_0 : \mathbb{X} \to \mathbb{R}^{d_0}, \tag{$(\cdot)_0$-gate}$$

$$\omega : \mathbb{X} \to C_{1,0}([0,1]; \mathbb{R}^{d_\omega}), \tag{$\omega$-gate}$$

$$\xi : \mathbb{X} \to C_{1,0}([0,1]; \mathbb{R}^{d_\xi}). \tag{$\xi$-gate}$$

where all the gates are continuous functions on $\mathbb{X}$. The space $\mathbb{X}$ does not have to be a space of paths, a topological structure suffices, as long as the gates $(\cdot)_0, \omega, \xi$ are well defined and continuous.

*Remark* B.1.  Typical examples for the choice of gates are $\mathbb{X}$ space of paths and

$$(X)_0 = 0 \quad \omega_t^X = t \quad \xi_t^X = \int_0^t X_s ds \tag{S4}$$

$$(X)_0 = 0 \quad \omega_t^X = \int_0^t softplus(\alpha X_s + \beta) ds \quad \xi_t^X = \int_0^t softplus(\alpha X_s + \beta) X_s ds \quad \text{(Mamba)}$$

Then the main object of study, "gated" Linear CDEs, are defined as:

**Definition B.2.**  Fix gates $(\cdot)_0, \omega, \xi$ as above, $N \in \mathbb{N}$, matrices $\{A_i\}_{i=1,\ldots,d_\omega}$ ($A_i \in \mathbb{R}^{N \times N}$), $B \in \mathbb{R}^{N \times d_\xi}, C \in \mathbb{R}^{N \times d_0}$. The corresponding Linear CDE is the functional

$$Z : \mathbb{X} \to C_1([0,1]; \mathbb{R}^N) \tag{24}$$

$$Z_0^X = CX_0, \quad Z_t^X = \sum_{i=1}^{d_\omega} A_i Z_t^X d\omega_t^{X,i} + Bd\xi_t^X \tag{25}$$

### B.2  Main Result - Statement and Strategy

Here we present the unified expressivity result in its most general form:

**Theorem B.3.**  *For any compact set $\mathbb{K} \subseteq \mathbb{X}$ and continuous gates $(\cdot)_0, \omega, \xi$ with $\omega_t^{X,1} \equiv t$ and $\omega_t^{X,2} \equiv t^2$. For any $\epsilon > 0$ and any*

$$F \in \left\{ (X,t) \mapsto \Psi(\omega_{[0,t]}^X) \cdot X_0 + \int_0^t \Phi(\omega_{[s,t]}^X) \cdot d\xi_s^X \right\} \tag{26}$$

*where $\Psi \in C^0(C_{1,0}; \mathbb{R}^{d_0})$ and $\Phi \in C^0(C_{1,0}; \mathbb{R}^{d_\xi})$, there exist a choice of hidden dimension $N \geq 1$ and parameters $v \in \mathbb{R}^N, A_i \in \mathbb{R}^{N \times N}, B \in \mathbb{R}^{N \times d_\xi}, C \in \mathbb{R}^{N \times d_0}$ such that*

$$\sup_{(X,t) \in \mathbb{K} \times [0,1]} |F(X,t) - \langle v, Z_t^X \rangle| \leq \epsilon \tag{27}$$

*Moreover generic parameters suffice with high probability in the sense that under LeCun initialization*

$$[A_i]_{n,j} \overset{iid}{\sim} \mathcal{N}(0, \frac{1}{N}) \quad C_{n,j}, B_{n,j} \overset{iid}{\sim} \mathcal{N}(0,1)$$

*the following holds:*

$$\lim_{N \to \infty} \mathbb{P}\big[\exists v \in \mathbb{R}^N \, : \, \text{(27) holds}\big] = 1$$

*If the $A_i$s are constrained to be diagonal, as often is the case in practice, the requirements $\omega_t^{X,1} \equiv t$, $\omega_t^{X,2} \equiv t^2$ can be dropped and the existence result only holds with*

$$F \in \left\{ (X,t) \mapsto \psi(\omega_t^X) \cdot X_0 + \int_0^t \phi(\omega_t^X - \omega_s^X) \cdot d\xi_s^X \right\} \tag{28}$$

*for $\psi \in C^0(\mathbb{R}^{d_\omega}; \mathbb{R}^{d_0})$ and $\phi \in C^0(\mathbb{R}^{d_\omega}; \mathbb{R}^{d_\xi})$.*

*Moreover in both the dense and diagonal cases the "reverse" also holds in the sense that, given any choice of matrices $A_i, B, C$ there is an $\epsilon$-close map $F$ in the corresponding family.*

As one can see the theorem is composed of different sub-results, which we believe are better understood separately from each other. The proof will thus be split in the following steps:

1. Using the theory developed in Appendix E we see how linear functions on the $Z_t$s can be seen as linear functions on certain terms of the Signature Transform.

2. Such terms define a *feature map* $T(X)_{0,t}$ which generates a Reproducing Kernel Hilbert Space $\mathcal{H}_t^{(\cdot)_0, \omega, \eta}$. This abstract space acts as an upper bound on expressivity: linear functions on $Z_t$ always belong to its closure (in uniform norm), independently of dimension and weights chosen, hence they cannot reach what functions in $\mathcal{H}_t^{(\cdot)_0, \omega, \eta}$ can't approximate.

3. The full expressive range of $\mathcal{H}_t^{(\cdot)_0, \omega, \eta}$ is shown to be captured by generic $Z_t$s.

4. Diagonal systems are shown to be restricted to a subset of the $\mathcal{H}_t^{(\cdot)_0, \omega, \eta}$ of which they capture the full expressive range.

### B.3 Main Result - Proofs

#### B.3.1 An expansion for $Z_t^{\mathbf{X}}$

**Proposition B.4.** *For any choice of $A_i \in \mathbb{R}^{N \times N}, B \in \mathbb{R}^{N \times d_\xi}$ and $C \in \mathbb{R}^{N \times d_0}$, the unique solution to*

$$dZ_t^X = \sum_{i=1}^{d_\omega} A_i Z_t^X d\omega_t^{X,i} + B d\xi_t^X \tag{29}$$

$$Z_0^X = CX_0 \in \mathbb{R}^N$$

*is given, using the notation $A_{Ij} := A_j A_I$, by*

$$Z_t^X = \sum_{i=1}^{d_0} \sum_{I \in \mathbb{W}_{d_\omega}} A_I C_i \, X_0^i Sig(\omega^X)_{0,t}^I + \sum_{j=1}^{d_\xi} \sum_{I \in \mathbb{W}_{d_\omega}} A_I B_j \int_0^t Sig(\omega^X)_{s,t}^I d\xi_s^{X,j} \in \mathbb{R}^N \tag{30}$$

*Notice here $A_I C_i, A_I B_j \in \mathbb{R}^N$ and $X_0^i Sig(\omega^X)_{0,t}^I, \int_0^t Sig(\omega^X)_{s,t}^I d\xi_s^{X,j} \in \mathbb{R}$.*

*Proof.* Just apply Theorems (E.2) and (E.6) of Appendix E. $\qquad\square$

*Remark* B.5. A property highlighted by the previous result is the interpretability of these models. After training the CDEs one can compute the matrix multiplications and observe which entries of the signature the model chooses to take into consideration, to attend.

#### B.3.2 The feature map $T$ and its RKHS

The expression in (30) is a *linear map* on a *feature vector* given, at time $t$, by

$$T(X)_{0,t} := \left( X_0^i Sig(\omega^X)_{0,t}^I, \int_0^t Sig(\omega^X)_{s,t}^I d\xi_s^{X,j} : i \in [d_0], I \in \mathbb{W}_{d_\omega}, j \in [d_\xi] \right) \tag{31}$$

This feature vector can be understood as a tensor in the following way:

**Definition B.6.** Let $\mathbb{W}_{d_0,d_\omega,d_\xi}$ be the set of words in the alphabet

$$\mathcal{A}_{d_0,d_\omega,d_\xi} := \{\boldsymbol{e}_i\}_{i=1,\ldots,d_0} \cup \{\boldsymbol{\epsilon}_j^\xi\}_{j=1,\ldots,d_\xi} \cup \{\boldsymbol{\epsilon}_k^\omega\}_{k=1,\ldots,d_\omega}$$

Fixed the gates $(\cdot)_0, \omega, \xi$ we define $T(X) : [0,1] \to l^2(\mathbb{W}_{d_0,d_\omega,d_\xi}) \subseteq T((\mathcal{A}_{d_0,d_\omega,d_\xi}))$ as the unique solution to:

$$T(X)_{0,t} = \sum_{i=1}^{d} X_0^i \boldsymbol{e}_i + \sum_{j=1}^{d_\xi} \xi_t^{X,j} \boldsymbol{\epsilon}_j^\xi + \sum_{k=1}^{d_\omega} \int_0^t T(X)_{0,s} \, d\omega_s^{X,k} \otimes \boldsymbol{\epsilon}_k^\omega \tag{32}$$

In fact one readily sees that the only non-zero terms of $T(X)_{0,t}$ defined as above are

$$\langle T(X)_{0,t}, \boldsymbol{e}_i \rangle = X_0^i$$

$$\langle T(X)_{0,t}, \boldsymbol{e}_i \otimes \boldsymbol{\epsilon}_{Ik}^\omega \rangle = \int_0^t \langle T(X)_{0,s}, \boldsymbol{e}_i \otimes \boldsymbol{\epsilon}_I^\omega \rangle d\omega_s^{X,k} = X_0^i \, \text{Sig}(\omega^X)_{0,t}^{Ik}$$

$$\langle T(X)_{0,t}, \boldsymbol{\epsilon}_j^\xi \rangle = \xi_t^{X,j} = \int_0^t d\xi_s^{X,j}$$

$$\langle T(X)_{0,t}, \boldsymbol{\epsilon}_j^\xi \otimes \boldsymbol{\epsilon}_{Ik}^\omega \rangle = \int_0^t \langle T(X)_{0,s}, \boldsymbol{\epsilon}_j^\xi \otimes \boldsymbol{\epsilon}_I^\omega \rangle d\omega_s^{X,k} = \int_{s=0}^t \int_{r=0}^s \text{Sig}(\omega^X)_{r,s}^I d\xi_r^{X,j} d\omega_s^{X,k}$$

$$= \int_{r=0}^t \int_{s=r}^t \text{Sig}(\omega^X)_{r,s}^I d\omega_s^{X,k} d\xi_r^{X,j} = \int_0^t \text{Sig}(\omega^X)_{r,t}^I \, d\xi_r^{X,j}$$

This is similar to the tensor-valued CDE defining the signature as a tensor *i.e.* Salvi et al. [2021a]

$$\text{Sig}(\omega)_{0,t} = () + \int_0^t \text{Sig}(\omega)_{0,s} \otimes d\omega_s$$

with the addition of two terms to track $X_0$ and $\xi^X$. One could also understand $T(X)_{s,t}$ as a sub-tensor of

$$X_0 \otimes \text{Sig}((\omega^X, \xi^X))_{s,t}$$

but in doing this one would have to explicitly ignore most of the terms of this vector; the CDE (32) does exactly this, but implicitly. In any case the subtensor view shows that $T : \mathbb{X} \times [0,1]^2 \to l^2(\mathbb{W}_{d_0,d_\omega,d_\xi})$ is well defined and continuous.

To the feature map $T(\cdot)_{0,t}$ with values in the Hilbert space $l^2(\mathbb{W}_{d_0,d_\omega,d_\xi})$ is then associated a Reproducing Kernel Hilbert Space Berlinet and Thomas-Agnan [2011], where the Kernel is the one induced by the $l^2$ product, which we denote by

$$\mathcal{H}_t^{(\cdot)_0,\omega,\eta} \subseteq C^0(\mathbb{X}; \mathbb{R}) \tag{33}$$

Classical RKHS theory tells us that we can characterize its elements as:

**Proposition B.7.** *A map* $F(\cdot)_t : \mathbb{X} \to \mathbb{R}$ *is an element of* $\mathcal{H}_t^{(\cdot)_0,\omega,\eta}$ *if and only if it is of the form*

$$F(x)_t = \sum_{i=1}^{d_0} X_0^i \langle \alpha_i, \text{Sig}(\omega^X)_{0,t} \rangle + \sum_{j=1}^{d_\xi} \int_0^t \langle \beta_j, \text{Sig}(\omega^X)_{s,t} \rangle d\xi_s^{X,j} \tag{34}$$

*for* $\alpha_i, \beta_j \in l^2(\mathbb{W}_{d_\omega})$. *Moreover* $\|F(\cdot)_t\|_{\mathcal{H}_t^{(\cdot)_0,\omega,\eta}}^2$ *is equal to the minimal value of*

$$\sum_{i=1}^{d_0} \|\alpha_i\|_{l^2(\mathbb{W}_{d_\omega})}^2 + \sum_{j=1}^{d_\xi} \|\beta_j\|_{l^2(\mathbb{W}_{d_\omega})}^2$$

*taken over those* $\gamma, \beta$ *for which the above equality holds.*

*Signature kernels* Salvi et al. [2021a] are a class of *universal kernels* on sequential data which have received attention in recent years thanks to their efficiency in handling path-dependent problems Lemercier et al. [2021], Salvi et al. [2021b], Cochrane et al. [2021], Salvi et al. [2021c], Cirone et al. [2023], Issa et al. [2023], Pannier and Salvi [2024], Manten et al. [2024].

Just as signature kernels, the kernel associated to $T(X)_{0,t}$ can be explicitly written as the solution of a two-parameter CDE:

**Lemma B.8.** *Let $\mathcal{K}^{X,Y}(s,t) := \langle T(X)_{0,s}, T(Y)_{0,t}\rangle_{l^2}$ then*

$$\mathcal{K}^{X,Y}(s,t) = \langle X_0, Y_0\rangle + \langle \xi_s^X, \xi_t^Y\rangle + \int_{\eta=0}^{s}\int_{\tau=0}^{t}\mathcal{K}^{X,Y}(\eta,\tau)\langle d\omega_\eta^X, d\omega_\tau^Y\rangle \tag{35}$$

*or, directly in terms of Signature, also*

$$\mathcal{K}^{X,Y}(s,t) = \langle X_0, Y_0\rangle\langle Sig(\omega^X)_{0,s}, Sig(\omega^Y)_{0,t}\rangle + \int_{\eta=0}^{s}\int_{\tau=0}^{t}\langle Sig(\omega^X)_{\eta,s}, Sig(\omega^Y)_{\tau,t}\rangle\langle d\xi_\eta^X, d\xi_\tau^Y\rangle \tag{36}$$

*Proof.* The first expression follows immediately from (32) the second one by summing the products of $T(X)_{0,s}$'s and $T(Y)_{0,t}$'s entries given above. $\qquad\square$

**Definition B.9.** Define the space $\mathcal{H}_{[0,1]}^{(\cdot)_0,\omega,\eta} \subseteq C^0(\mathbb{X}\times[0,1];\mathbb{R})$ as the space of functions of form

$$(X,t) \mapsto \sum_{i=1}^{d_0} X_0^i\langle\alpha_i, \text{Sig}(\omega^X)_{0,t}\rangle + \sum_{j=1}^{d_\xi}\int_0^t\langle\beta_j, \text{Sig}(\omega^X)_{s,t}\rangle d\xi_s^{X,j} \tag{37}$$

for $\alpha_i, \beta_j \in l^2(\mathbb{W}_{d_\omega})$. Thus for all $t \in [0,1]$ and $F \in \mathcal{H}_{[0,1]}^{(\cdot)_0,\omega,\eta}$ it holds $F(\cdot,t) \in \mathcal{H}_t^{(\cdot)_0,\omega,\eta}$.

### B.3.3  Linear maps on $Z_t^{\mathbf{X}}$ are close to the RKHS

The following proposition will show how linear maps on $Z_t^{\mathbf{X}}$ cannot be more expressive than elements of the RKHS $\mathcal{H}_{[0,1]}^{(\cdot)_0,\omega,\eta}$ since their closure is in the closure of $\mathcal{H}_{[0,1]}^{(\cdot)_0,\omega,\eta}$. In this precise sense these spaces act like upper bounds to expressiveness.

**Proposition B.10.** *Assume $\mathbb{X}$ compact. Consider fixed the gates and $A_i \in \mathbb{R}^{N\times N}, B \in \mathbb{R}^{N\times d_\xi}$ and $C \in \mathbb{R}^{N\times d_0}$. Consider a linear readout $v \in \mathbb{R}^N$. For any $\epsilon > 0$ there exist choices of $\alpha_i, \beta_j \in l^2(\mathbb{W}_{d_\omega})$ such that*

$$\sup_{(X,t)\in\mathbb{X}\times[0,1]} |\langle v, Z_t^{\mathbf{X}}\rangle - F(X,t)| \leq \epsilon \tag{38}$$

*where $F \in \mathcal{H}_{[0,1]}^{(\cdot)_0,\omega,\eta}$. In other words, linear maps on the $Z_t^{\mathbf{X}}$ are in the uniform closure of $\mathcal{H}_{[0,1]}^{(\cdot)_0,\omega,\eta}$.*

*Proof.* Using (30) we see that $\langle v, Z_t^{\mathbf{X}}\rangle$ is a linear map on $T(X)_{0,t}$ with coefficients

$$v^\top A_I B_j \quad v^\top A_I C_i$$

using Cauchy-Schwartz it's moreover easy to see the existence of a constant $\lambda \geq 0$ such that for all $I, i, j$ one has

$$|v^\top A_I B_j| \leq \lambda^{|I|} \quad |v^\top A_I C_i| \leq \lambda^{|I|}.$$

Since $|\text{Sig}(\omega)_{s,t}^I| \leq \frac{1}{|I|!}\|\omega\|_{1-var,[s,t]}^{|I|}$ we have that, given an integer $M \geq 0$, the bound

$$R_M(t) := \left|\sum_{i=1}^{d_0}\sum_{|I|\geq M} v^\top A_I C_i\, X_0^i\text{Sig}(\omega^X)_{0,t}^I + \sum_{j=1}^{d_\xi}\sum_{|I|\geq M} v^\top A_I B_j\int_0^t\text{Sig}(\omega^X)_{s,t}^I d\xi_s^{X,j}\right|$$

$$\leq (\|X_0\|_1 + \|\xi_t^X\|_1)\sum_{m=M}^{\infty}\frac{\lambda^m d_\omega^m\|\omega^X\|_{1-var,[0,t]}^m}{m!}$$

$$\leq (\|X_0\|_1 + \|\xi_t^X\|_1)\sum_{m=M}^{\infty}\frac{\lambda^m d_\omega^m\|\omega^X\|_{1-var,[0,1]}^m}{m!} \leq K\sum_{m=M}^{\infty}\frac{(\lambda d_\omega K)^m}{m!}$$

where $K \geq 0$ is a constant which must exist by compactness of $\mathbb{X}$ and continuity of the gates. Since $K\sum_{m=M}^{\infty}\frac{(\lambda d_\omega K)^m}{m!}$ is just the tail of the taylor expansion of $Ke^{\lambda d_\omega K}$ there must be an $M$ such that $\sup_{t\in[0,1]} R_M(t) \leq \epsilon$. But then the choice

$$\alpha_i^I := v^\top A_I C_i\, \mathbb{I}(|I| < M) \quad \beta_j^I := v^\top A_I B_j\, \mathbb{I}(|I| < M)$$

suffices for the required bound.

$\qquad\square$

### B.3.4 Uniform closure of the RKHS

Now that we have established the theoretical interest of the $\mathcal{H}_{[0,1]}^{(\cdot)_0,\omega,\eta}$ we proceed to characterize which maps $(X,t) \to \mathbb{R}$ can be uniformly approximated through them.

**Proposition B.11.** *Fix a compact input set $\mathbb{X}$ and continuous gates $(\cdot)_0, \omega, \xi$ with $\omega_t^{X,1} \equiv t$ and $\omega_t^{X,2} \equiv t^2$. For any $\epsilon > 0$ and any*

$$F \in \left\{ (X,t) \mapsto \Psi(\omega_{[0,t]}^X) \cdot X_0 + \int_0^t \Phi(\omega_{[s,t]}^X) \cdot d\xi_s^X \right\} \tag{39}$$

*where $\Psi \in C^0(C_{1,0}; \mathbb{R}^{d_0})$ and $\Phi \in C^0(C_{1,0}; \mathbb{R}^{d_\xi})$, there exist a $G \in \mathcal{H}_{[0,1]}^{(\cdot)_0,\omega,\eta}$ such that*

$$\sup_{(X,t)\in\mathbb{X}\times[0,1]} |F(X,t) - G(X,t)| \leq \epsilon \tag{40}$$

*Proof.* Note first that the map

$$\mathbb{X} \times [0,1] \times [0,1] \to C_{1,0}([0,1]; \mathbb{R}^{d_\omega}) \quad (X,s,t) \mapsto \omega_{[s,t]}^X$$

is a continuous map from a compact space, thus the image must be compact too. Moreover by Prop. A.8 the Signature separates the points in this image. Since any $G$ as above has form

$$G(X,t) = \sum_{i=1}^{d_0} X_0^i \langle \alpha_i, \mathrm{Sig}(\omega^X)_{0,t} \rangle + \sum_{j=1}^{d_\xi} \int_0^t \langle \beta_j, \mathrm{Sig}(\omega^X)_{s,t} \rangle d\xi_s^{X,j}$$

$$= \sum_{i=1}^{d_0} X_0^i \langle \alpha_i, \mathrm{Sig}(\omega_{[0,t]}^X)_{0,1} \rangle + \sum_{j=1}^{d_\xi} \int_0^t \langle \beta_j, \mathrm{Sig}(\omega_{[s,t]}^X)_{0,1} \rangle d\xi_s^{X,j}$$

the proof follows from the uniform density on compact sets of linear functionals on the (truncated) Signature (Thm. A.5), by also uniformly bounding thanks to compactness and continuity the norms of $X_0$ and $\xi_1^X$. $\square$

*Remark* B.12. The specific restriction of $\omega$ to subsets of $[0,1]$ is a crucial part of the result. The family of approximable maps *does not* include *all* path-to-path causal[11] functions $t \mapsto Y_t^X$ but a subset of them, of type $t \mapsto Y_t^X := \Psi(\omega_{[0,t]}^X)$, satisfying the specific *time-homogeneity* specified by the form of the restriction, akin to that in Li et al. [2022b].

### B.3.5 Generic Weights are fully expressive

We have seen how linear maps on $Z_t^X$ are in the uniform closure of $\mathcal{H}_{[0,1]}^{(\cdot)_0,\omega,\eta}$, and we have explicitly characterized this closure. It is then natural to ask "how much" of this closure the $Z_t^X$ are able to "explore". The present section not only shows that the $Z_t^X$ "explore" all the closure, but also that a generic choice of weights is enough to eventually do this with high probability.

The fact that these maps are "universal" in the above sense is not surprising, since it is well known that Linear CDEs are universal for path-to-point tasks *cf.* Kidger [2022], what is surprising is that this universality can be achieved probabilistically with one of the *standard* parametrizations used in ML practice (LeCun) [12].

**Theorem B.13.** *Fix $\mathbb{X}$ compact and $\epsilon > 0$. For all $F \in \mathcal{H}_{[0,1]}^{(\cdot)_0,\omega,\eta}$ there exist a choice of hidden dimension $N \geq 1$ and parameters $v \in \mathbb{R}^N, A_i \in \mathbb{R}^{N\times N}, B \in \mathbb{R}^{N\times d_\xi}, C \in \mathbb{R}^{N\times d_0}$ such that*

$$\sup_{(X,t)\in\mathbb{X}\times[0,1]} |F(X,t) - \langle v, Z_t^X \rangle| \leq \epsilon \tag{41}$$

*Moreover generic weight choices suffice with high probability, in the sense that under LeCun initialization*

$$[A_j]_{n,n'} \overset{iid}{\sim} \mathcal{N}(0, \frac{1}{N}) \quad [C]_{n,i}, [B]_{n,j} \overset{iid}{\sim} \mathcal{N}(0,1)$$

---

[11]A *causal* map is one which does not "look in the future" *cf.* Appendix D.

[12]It can be proved, using the results of Dubach and Peled [2021], that the sampling measure does not have to be Gaussian if it satisfies certain moment requirements.

*the following holds*

$$\lim_{N\to\infty} \mathbb{P}\big[\exists v \in \mathbb{R}^N : \ (41)\ holds\big] = 1$$

We propose two proofs, the first one of a deterministic character concerns the first claim in the theorem, the second one is probabilistic and concerns the whole result. The deterministic proof follows the same arguments employed by Kidger [2022] and is included to highlight the main idea of the probabilistic result, which reduces to a "spin" on the central argument of this proof.

*Deterministic Proof.* Any $F \in \mathcal{H}_{[0,1]}^{(\cdot)_0,\omega,\eta}$ has form

$$F(X,t) = \sum_{i=1}^{d_0} X_0^i \langle \alpha_i, \mathrm{Sig}(\omega^{\mathrm{X}})_{0,t} \rangle + \sum_{j=1}^{d_\xi} \int_0^t \langle \beta_j, \mathrm{Sig}(\omega^{\mathrm{X}})_{s,t} \rangle d\xi_s^{\mathrm{X},j} \tag{42}$$

for fixed $\alpha_i, \beta_j \in l^2(\mathbb{W}_{d_\omega})$. Consider an integer $M \geq 0$ such that

$$\sup_{(x,t) \in \mathbb{X} \times [0,1]} |F(X,t) - \sum_{i=1}^{d_0} X_0^i \langle \pi_M \alpha_i, \mathrm{Sig}(\omega^{\mathrm{X}})_{0,t} \rangle - \sum_{j=1}^{d_\xi} \int_0^t \langle \pi_M \beta_j, \mathrm{Sig}(\omega^{\mathrm{X}})_{s,t} \rangle d\xi_s^{\mathrm{X},j}| \leq \epsilon \tag{43}$$

where $\pi_M$ is the truncation at length $M$.

Fix $d = d_0 + d_\omega + d_\xi$. Consider $\mu(M,d) \in \mathbb{N}$ such that $\mathbb{R}^{\mu(M,d)} \simeq T^M(\mathbb{R}^d)$. We are going to write $e_I \in \mathbb{R}^{\mu(M,d)}$ to mean the image of $e_I \in T^M(\mathbb{R}^d)$ through this identification. Note that $(\cdot) \otimes_M e_k : T^M(\mathbb{R}^d) \to T^M(\mathbb{R}^d)$ is a linear map, it does then correspond to a matrix $\Lambda_k \in \mathbb{R}^{\mu(M,d) \times \mu(M,d)}$. Write $\varepsilon_j^\xi := e_{d_0+j}$ for $j = 1, \ldots, d_\xi$ and $\varepsilon_k^\omega := e_{d_0+d_\xi+k}$ for $k = 1, \ldots, d_\omega$. Then the solution to

$$\mathbb{R}^{\mu(M,d)} \ni \tilde{Z}_t = \sum_{i=1}^{d_0} X_0^i e_i + \sum_{j=1}^{d_\xi} \xi_t^{\mathrm{X},j} \varepsilon_j^B + \sum_{k=1}^{d_\omega} \int_0^t \Lambda_{d_0+d_\xi+k} \tilde{Z}_s d\omega_s^{\mathrm{X},k} \tag{44}$$

is the object in $\mathbb{R}^{\mu(M,d)}$ corresponding to the truncated tensor $\pi_M(T(X)_{0,t})$.

This $\tilde{Z}_t$ is of the form $Z_t^{\mathrm{X}}$ with $N = \mu(M,d)$, $A_k = \Lambda_{d_0+d_\xi+k}$, $B = \left[\varepsilon_1^\xi | \cdots | \varepsilon_{d_\xi}^\xi\right]$ and $C = [e_1 | \cdots | e_{d_0}]$.

In particular note how these matrices are such that

$$e_J^T A_I C_i = \mathbb{I}(e_J = e_i \otimes \varepsilon_I^\omega), \quad e_J^T A_I B_j = \mathbb{I}(e_J = \varepsilon_j^\xi \otimes \varepsilon_I^\omega), \tag{45}$$

since it holds that

$$A_I C_i = e_i \otimes \varepsilon_I^\omega, \quad A_I B_j = \varepsilon_j^\xi \otimes \varepsilon_I^\omega, \tag{46}$$

and for all $|I| > M$ one has necessarily $A_I = 0$.

Our strategy is that of using these equaities to create a vector $v \in \mathbb{R}^N$ corresponding to the $\pi_M \alpha_i$ and $\pi_M \beta_j$. Define the vector

$$v := \sum_{i=1}^{d_0} \sum_{|I| \leq M} \alpha_i^I \ e_i \otimes \varepsilon_I^\omega + \sum_{j=1}^{d_\omega} \sum_{|I| \leq M} \beta_j^I \ \varepsilon_j^\xi \otimes \varepsilon_I^\omega \in \mathbb{R}^{\mu(M,d)} \tag{47}$$

Then expanding $Z_t^{\mathrm{X}}$ as in (30) and using the equalities above one has

$$\begin{aligned}
\langle v, Z_t^{\mathrm{X}} \rangle &= \sum_{i=1}^{d_0} \sum_I v^\top A_I C_i \ X_0^i \mathrm{Sig}(\omega^{\mathrm{X}})_{0,t} + \sum_{j=1}^{d_\omega} \sum_I v^\top A_I B_j \ \int_0^t \mathrm{Sig}(\omega^{\mathrm{X}})_{s,t} d\xi_s^{\mathrm{X},j} \\
&= \sum_{i=1}^{d_0} \sum_{|I| \leq M} \alpha_i^I X_0^i \mathrm{Sig}(\omega^{\mathrm{X}})_{0,t} + \sum_{j=1}^{d_\xi} \sum_{|I| \leq M} \beta_j^I \int_0^t \mathrm{Sig}(\omega^{\mathrm{X}})_{s,t} d\xi_s^{\mathrm{X},j} \\
&= \sum_{i=1}^{d_0} X_0^i \langle \pi_M \alpha_i, \mathrm{Sig}(\omega^{\mathrm{X}})_{0,t} \rangle + \sum_{j=1}^{d_\xi} \int_0^t \langle \pi_M \beta_j, \mathrm{Sig}(\omega^{\mathrm{X}})_{s,t} \rangle d\xi_s^{\mathrm{X},j}
\end{aligned}$$

proving that for such $v$ and $Z^X$ it holds

$$\sup_{(x,t)\in\mathbb{X}\times[0,1]} |F(X,t) - \langle v, Z_t^X\rangle| \leq \epsilon \tag{48}$$

$\square$

The crucial ingredient for the success of this proof is the possibility to recreate the space $T^M(\mathbb{R}^{d_0+d_\omega+d_\xi})$ as an euclidean space. To do this one needs $\mu(M, d_0+d_\omega+d_\xi) \sim (d_0+d_\omega+d_\xi)^M$ orthogonal vectors and a way to express them using the matrices $A_i, B$ and $C$, the essential equations which capture this are given by (45).

The core idea of the following probabilistic proof of this same result is that of allowing for some error in (45), so the idea is that of exhibiting only *approximately* orthogonal vectors. At the cost of losing exactness, one can leverage results of the Johnson-Lindenstrauss Dasgupta and Gupta [2003] type to find on the order of $\sim e^{\varepsilon^2 N}$ vectors in $\mathbb{R}^N$ orthogonal up to an $\varepsilon$ error, using random projections. This idea in the context of Signature goes back to Cuchiero et al. [2021b], and allows for much smaller hidden dimensions.

*Proof.* (Probabilistic Proof) Any $F \in \mathcal{H}_{[0,1]}^{(\cdot)_0,\omega,\eta}$ has form

$$F(X,t) = \sum_{i=1}^{d_0} X_0^i \langle \alpha_i, \mathrm{Sig}(\omega^X)_{0,t}\rangle + \sum_{j=1}^{d_\xi} \int_0^t \langle \beta_j, \mathrm{Sig}(\omega^X)_{s,t}\rangle d\xi_s^{X,j} \tag{49}$$

for fixed $\alpha_i, \beta_j \in l^2(\mathbb{W}_{d_\omega})$. Consider an integer $M \geq 0$ such that

$$\sup_{(x,t)\in\mathbb{X}\times[0,1]} |F(X,t) - \sum_{i=1}^{d_0} X_0^i \langle \pi_M\alpha_i, \mathrm{Sig}(\omega^X)_{0,t}\rangle - \sum_{j=1}^{d_\xi} \int_0^t \langle \pi_M\beta_j, \mathrm{Sig}(\omega^X)_{s,t}\rangle d\xi_s^{X,j}| \leq \epsilon \tag{50}$$

where $\pi_M$ is the truncation at length $M$.

From Cirone et al. [2023][Appendix C] we know that

$$\left\|\frac{1}{N}C_i^\top A_I^\top A_J C_j - \delta_{iI}^{jJ}\right\|_{L^2} = \mathcal{O}(\frac{1}{\sqrt{N}})2^{\frac{|I|+|J|}{2}}(|I|+|J|)!! \tag{51}$$

$$\left\|\frac{1}{N}C_i^\top A_I^\top A_J B_j\right\|_{L^2} = \mathcal{O}(\frac{1}{\sqrt{N}})2^{\frac{|I|+|J|}{2}}(|I|+|J|)!! \tag{52}$$

$$\left\|\frac{1}{N}B_i^\top A_I^\top A_J B_j - \delta_{iI}^{jJ}\right\|_{L^2} = \mathcal{O}(\frac{1}{\sqrt{N}})2^{\frac{|I|+|J|}{2}}(|I|+|J|)!! \tag{53}$$

Our strategy is that of using these bounds to create a vector $v \in \mathbb{R}^N$ "acting" like the $\pi_M\alpha_i$ and $\pi_M\beta_j$. Define, noting that the $A_i, C_i, B_j$ depend on $N$, the vector

$$v^N := \frac{1}{N}\left(\sum_{i=1}^{d_0}\sum_{|I|\leq M} \alpha_i^I A_I C_i + \sum_{j=1}^{d_\omega}\sum_{|I|\leq M} \beta_j^I A_I B_j\right) \tag{54}$$

Then expanding $Z_t^{\mathrm{X}}$ as in (30)

$$
R_M := \left\| \sup_{(X,t)\in\mathbb{X}\times[0,1]} \left| \langle v^{\mathrm{N}}, Z_t^{\mathrm{X}} \rangle - \sum_{i=1}^{d_0} X_0^i \langle \pi_M \alpha_i, \mathrm{Sig}(\omega^{\mathrm{X}})_{0,t} \rangle - \sum_{j=1}^{d_\xi} \int_0^t \langle \pi_M \beta_j, \mathrm{Sig}(\omega^{\mathrm{X}})_{s,t} \rangle d\xi_s^{\mathrm{X},j} \right| \right\|_{L^2}
$$

$$
\leq \sum_{i=1}^{d_0} \sum_{|I|\leq M} \left\| (v^{\mathrm{N}})^\top A_I C_i - \alpha_i^I \right\|_{L^2} \sup_{(X,t)\in\mathbb{X}\times[0,1]} |X_0^i \mathrm{Sig}(\omega^{\mathrm{X}})_{0,t}^I|
$$

$$
+ + \sum_{i=1}^{d_0} \sum_{|I|>M} \left\| (v^{\mathrm{N}})^\top A_I C_i \right\|_{L^2} \sup_{(X,t)\in\mathbb{X}\times[0,1]} |X_0^i \mathrm{Sig}(\omega^{\mathrm{X}})_{0,t}^I|
$$

$$
+ \sum_{j=1}^{d_\omega} \sum_{|I|\leq M} \left\| (v^{\mathrm{N}})^\top A_I B_j - \beta_j^I \right\|_{L^2} \sup_{(X,t)\in\mathbb{X}\times[0,1]} |\int_0^t Sig(\omega^{\mathrm{X}})_{s,t}^I d\xi_s^{\mathrm{X},j}|
$$

$$
+ + \sum_{j=1}^{d_\omega} \sum_{|I|>M} \left\| (v^{\mathrm{N}})^\top A_I B_j \right\|_{L^2} \sup_{(X,t)\in\mathbb{X}\times[0,1]} |\int_0^t Sig(\omega^{\mathrm{X}})_{s,t}^I d\xi_s^{\mathrm{X},j}|
$$

Note how for $|I| \leq M$ one has

$$
\left\| (v^{\mathrm{N}})^\top A_I C_i - \alpha_i^I \right\|_{L^2} \leq \mathcal{O}_M(\frac{1}{\sqrt{N}})
$$

and that similarly for $|I| > M$

$$
\left\| (v^{\mathrm{N}})^\top A_I C_i \right\|_{L^2} \leq \mathcal{O}_M(\frac{1}{\sqrt{N}}) 2^{\frac{|I|}{2}} (M + |I|)!!
$$

Which leads, thanks to the same bounds of Cirone et al. [2023][Appendix C], to

$$
R_M = \frac{1}{\sqrt{N}} \mathcal{O}_{M,\mathbb{X}}(1) \tag{55}
$$

But then by Markov's inequality it holds that

$$
\mathbb{P}\left[ \sup_{(X,t)\in\mathbb{X}\times[0,1]} \left| \langle v^{\mathrm{N}}, Z_t^{\mathrm{X}} \rangle - \sum_{i=1}^{d_0} X_0^i \langle \pi_M \alpha_i, \mathrm{Sig}(\omega^{\mathrm{X}})_{0,t} \rangle - \sum_{j=1}^{d_\xi} \int_0^t \langle \pi_M \beta_j, \mathrm{Sig}(\omega^{\mathrm{X}})_{s,t} \rangle d\xi_s^{\mathrm{X},j} \right| \leq \epsilon \right] \to 1 \tag{56}
$$

and thus there must be a choice of $N, \{A_i\}, B, C$ such that the inequality holds, and we thus obtain using (50)

$$
\sup_{(X,t)\in\mathbb{X}\times[0,1]} |F(X,t) - \langle v^{\mathrm{N}}, Z_t^{\mathrm{X}} \rangle| \leq 2\epsilon
$$

and we conclude by arbitrariness of $\epsilon$.

$\square$

## B.4 The Diagonal Case

Here we study the particular, but empirically important, case where the matrices $A_i$ are taken to be diagonal[13].

What we'll discover is that the $Z_t^{\mathrm{X}}$ cannot differentiate between $\mathrm{Sig}(\omega^{\mathrm{X}})_{s,t}^I$ and other $\mathrm{Sig}(\omega^{\mathrm{X}})_{s,t}^{\sigma(I)}$ for any permutation $\sigma$ of the letters in the word $I$.

---

[13]It is equivalent to ask for them to be commuting.

### B.4.1 Diagonal Expansion for $Z_t^X$

**Proposition B.14.** *For any choice of $V \in \mathbb{R}^{N \times d_\omega}, B \in \mathbb{R}^{N \times d_\xi}$ and $C \in \mathbb{R}^{N \times d_0}$, writing $A_i := \mathrm{diag}(V_i)$, the unique solution to*

$$dZ_t^X = \sum_{i=1}^{d_\omega} A_i Z_t^X d\omega_t^{X,i} + B d\xi_t^X \tag{57}$$

$$Z_0^X = C X_0 \in \mathbb{R}^N$$

*is given by*

$$Z_t^X = e^{\mathrm{diag}(V \omega_t^X)} C X_0 + \int_0^t e^{\mathrm{diag}(V(\omega_t^X - \omega_s^X))} B d\xi_s^X \tag{58}$$

*which can be expanded as*

$$Z_t^X = \sum_{i=1}^{d_0} \sum_{I \in \mathbb{W}_{d_\omega}} A_I^{sym} C_i \, X_0^i \mathrm{Sig}(\omega^X)_{0,t}^{sym,I} + \sum_{j=1}^{d_\xi} \sum_{I \in \mathbb{W}_{d_\omega}} A_I^{sym} B_j \int_0^t \mathrm{Sig}(\omega^X)_{s,t}^{sym,I} d\xi_s^{X,j} \in \mathbb{R}^N \tag{59}$$

*where*

$$A_I^{sym} := \frac{1}{|I|!} \sum_{\sigma \in S_k} A_{\sigma(I)} = A_I \quad \mathrm{Sig}(\omega^X)_{s,t}^{sym,I} := \frac{1}{|I|!} \sum_{\sigma \in S_k} \mathrm{Sig}(\omega^X)_{s,t}^{\sigma(I)}. \tag{60}$$

*Proof.* By Theorem E.1 and Theorem E.6 we know that the solution of

$$Z_t^X = Z_0^X + \sum_{i=1}^{d_\omega} \int_0^t A_i Z_t^X d\omega_t^{X,i} + \int_0^t B d\xi_t^X \tag{61}$$

is explicitly given by

$$Z_t^X = W_{0,t}^X Z_0^X + \int_0^t W_{s,t}^X B d\xi_s^X \tag{62}$$

where $W_{s,t}$ is the unique solution to

$$W_{s,t}^X = Id + \sum_{i=1}^{d_\omega} \int_s^t A_i W_{s,r}^X d\omega_r^{X,i} \tag{63}$$

In case the $A_i$s are commuting matrices one can explicitly write the solution as

$$W_{s,t}^X = \exp\left( \sum_{i=1}^{d_\omega} \int_s^t A_i d\omega_r^{X,i} \right) = \exp\left( \mathrm{diag}(V(\omega_t^X - \omega_s^X)) \right) \tag{64}$$

since for fixed $s$ one has, using commutativity, that

$$dW_{s,t}^X = W_{s,t}^X \left( \sum_{i=1}^{d_\omega} A_i d\omega_t^{X,i} \right) = \sum_{i=1}^{d_\omega} A_i W_{s,t}^X d\omega_t^{X,i}$$

On the other hand we know, Theorem E.2, that

$$W_{s,t}^X = \sum_{I \in \mathbb{W}_{d_\omega}} A_I \mathrm{Sig}(\omega^X)_{s,t}^I = \sum_{k=0}^{\infty} \sum_{I \in \mathbb{W}_{d_\omega}^k} A_I \mathrm{Sig}(\omega^X)_{s,t}^I \tag{65}$$

The two views are reconciled by noticing that the symmetric group $S_k$ acts on $\mathbb{W}_{d_\omega}^k$, the space of words of lenth $k$, by permuting the letters and, by commutativity,

$$\forall \sigma \in S_k. \forall I \in \mathbb{W}_{d_\omega}^k. \ A_I = A_{\sigma(I)}$$

Then we have

$$\sum_{I \in \mathbb{W}_{d_\omega}^k} A_I \mathrm{Sig}(\omega^X)_{s,t}^I = \sum_{I \in \mathbb{W}_{d_\omega}^k} \frac{1}{k!} \sum_{\sigma \in S_k} A_{\sigma(I)} \mathrm{Sig}(\omega^X)_{s,t}^{\sigma(I)} = \sum_{I \in \mathbb{W}_{d_\omega}^k} \frac{A_I}{k!} \sum_{\sigma \in S_k} \mathrm{Sig}(\omega^X)_{s,t}^{\sigma(I)}$$

recalling then how $e_{I_1} \sqcup\cdots\sqcup e_{I_k} = \sum_{\sigma\in S_k} e_{\sigma(I)}$ we get to

$$\sum_{I\in\mathbb{W}^k_{d_\omega}} A_I \mathrm{Sig}(\omega^{\mathrm{X}})^I_{s,t}$$

$$= \sum_{I\in\mathbb{W}^k_{d_\omega}} \frac{A_I}{k!} \sum_{\sigma\in S_k} \mathrm{Sig}(\omega^{\mathrm{X}})^{\sigma(I)}_{s,t} = \sum_{I\in\mathbb{W}^k_{d_\omega}} \frac{A_I}{k!} \prod_{i=1}^k \mathrm{Sig}(\omega^{\mathrm{X}})^{I_i}_{s,t}$$

$$= \sum_{I\in\mathbb{W}^k_{d_\omega}} \frac{1}{k!} \prod_{i=1}^k A_{I_i} \mathrm{Sig}(\omega^{\mathrm{X}})^{I_i}_{s,t} = \frac{1}{k!} \left(\sum_{i=1}^{d_\omega} A_i \mathrm{Sig}(\omega^{\mathrm{X}})^i_{s,t}\right)^k = \frac{1}{k!} \left(\sum_{i=1}^{d_\omega} \int_s^t A_i d\omega^{\mathrm{X},i}_r\right)^k$$

In particular we see how in the commuting case

$$W^{\mathrm{X}}_{s,t} = \sum_{I\in\mathbb{W}_{d_\omega}} A^{sym}_I \mathrm{Sig}(\omega^{\mathrm{X}})^{sym,I}_{s,t} \tag{66}$$

where

$$A^{sym}_I := \frac{1}{|I|!} \sum_{\sigma\in S_k} A_{\sigma(I)} = A_I \quad \mathrm{Sig}(\omega^{\mathrm{X}})^{sym,I}_{s,t} := \frac{1}{|I|!} \sum_{\sigma\in S_k} \mathrm{Sig}(\omega^{\mathrm{X}})^{\sigma(I)}_{s,t}.$$

$\square$

### B.4.2 Diagonal Expressiveness

**Theorem B.15.** *Fix a compact input set $\mathbb{X}$ and continuous gates $(\cdot)_0, \omega, \xi$. For any $\epsilon > 0$ and any*

$$F \in \left\{ (X,t) \mapsto \psi(\omega^X_t)\cdot X_0 + \int_0^t \phi(\omega^X_t - \omega^X_s)\cdot d\xi^X_s \right\} \tag{67}$$

*for $\psi \in C^0(\mathbb{R}^{d_\omega}; \mathbb{R}^{d_0})$ and $\phi \in C^0(\mathbb{R}^{d_\omega}; \mathbb{R}^{d_\xi})$, there exist a choice of hidden dimension $N \geq 1$ and parameters $v \in \mathbb{R}^N, B \in \mathbb{R}^{N\times d_\xi}, C \in \mathbb{R}^{N\times d_0}$ and diagonal $A_i \in \mathbb{R}^{N\times N}$ such that*

$$\sup_{(X,t)\in\mathbb{X}\times[0,1]} |F(X,t) - \langle v, Z^X_t\rangle| \leq \epsilon \tag{68}$$

*Moreover the "reverse" also holds i.e. given any choice of matrices $A_i, B, C$ there is an $\epsilon$-close map $F$ in the family.*

*Proof.* This is just a repetition of the arguments used for the dense case with little more care to get the uniformity in time.

One defines the subset $Sym(\mathcal{H}^{(\cdot)_0,\omega,\eta}_{[0,1]}) \subset \mathcal{H}^{(\cdot)_0,\omega,\eta}_{[0,1]}$ of those $F$ of type (37) defined by $\alpha_i, \beta_j \in l^2(\mathbb{W}_{d_\omega})$ such that for any word $I$ and any permutation $\sigma(I)$ of it

$$\alpha^I_i = \alpha^{\sigma(I)}_i \quad \beta^I_j = \beta^{\sigma(I)}_j \tag{69}$$

The same argument of Proposition B.10 shows that the uniform closure of the space of linear maps on the $Z^X_t$ is contained in the uniform closure of $Sym(\mathcal{H}^{(\cdot)_0,\omega,\eta}_{[0,1]})$, and the same bounds show that this latter closure is the same as that of its subset composed of those $F \in Sym(\mathcal{H}^{(\cdot)_0,\omega,\eta}_{[0,1]})$ having entries eventually equal to 0.

Since

$$\mathrm{Sig}(\omega^{\mathrm{X}})^{sym,I}_{s,t} := \frac{1}{|I|!} \sum_{\sigma\in S_k} \mathrm{Sig}(\omega^{\mathrm{X}})^{\sigma(I)}_{s,t} = \frac{1}{|I|!} \prod_{i=1}^{|I|} (\omega^{\mathrm{X},I_i}_t - \omega^{\mathrm{X},I_i}_s),$$

such maps can be expressed exactly in the form

$$P(\omega^{\mathrm{X}}_t)\cdot X_0 + \int_0^t Q(\omega^{\mathrm{X}}_t - \omega^{\mathrm{X}}_s)\cdot d\xi^{\mathrm{X}}_s$$

for polynomial maps $P, Q$ fixed in time. The usual compactness and continuity argument, together with an application of Stone-Weiestrass, thus proves that the uniform closure of $Sym(\mathcal{H}_{[0,1]}^{(\cdot)_0,\omega,\eta})$ has the form needed.

The final ingredient is the density of the space of linear maps on the $Z_t^{\mathrm{X}}$ in $Sym(\mathcal{H}_{[0,1]}^{(\cdot)_0,\omega,\eta})$; this is another consequence of Stone-Weiestrass as seen from Proposition B.18. $\qquad\square$

*Remark* B.16. Notice how here there is no need to augment the paths in creative ways in order to ensure separability of the points. The map $(\omega, s, t) \mapsto \omega_{[s,t]} \in C_{1,0}([0,1]; \mathbb{R}^{d_\omega})$ is replaced by $(\omega, s, t) \mapsto \omega_t - \omega_s \in \mathbb{R}^{d_\omega}$ and the space of polynomials always separates points in $\mathbb{R}^{d_\omega}$.

*Remark* B.17. It is not necessary to pass through $Sym(\mathcal{H}_{[0,1]}^{(\cdot)_0,\omega,\eta})$ to prove the previous result, since it directly follows from Proposition B.18. This choice of presentation has been motivated by the conviction of the usefulness of drawing parallels and comparisons.

**Proposition B.18.** *Fix a compact set $\mathbb{K} \subset \mathbb{R}^d$ and a $d$-dimensional convex cone $C$ containing the origin. The space*

$$\mathcal{E} := Span\left(\mathbb{K} \ni x \mapsto e^{\langle \alpha, x \rangle_{\mathbb{R}^d}} \in \mathbb{R} : \alpha \in C\right)$$

*is uniformly dense in $C^0(\mathbb{K}; \mathbb{R})$.*

*Proof.* This is an application of Stone-Weiestrass: $\mathcal{E}$ is a sub-algebra since

$$e^{\langle \alpha, x \rangle} e^{\langle \beta, x \rangle_{\mathbb{R}^d}} = e^{\langle \alpha + \beta, x \rangle_{\mathbb{R}^d}}$$

and $\alpha, \beta \in C \implies \alpha + \beta \in C$ by convexity of the cone; $\mathcal{E}$ contains the constant function $e^{\langle 0, x \rangle} = 1$ and is clearly point separating since the cone, being $d$-dimensional, it contains a basis of the whole space. $\qquad\square$

*Remark* B.19. The usefulness of stating the previous result in such a general setting is the following: with this formalism we can, for example, restrict to $\alpha \le 0$, in this way we would have a method to control the stability (*cf.* Appendix C.1) of the Linear CDEs by choosing the gate with a.s. $\dot{\omega}^{\mathrm{X}} \ge 0$.

**Corollary B.20** (Mamba Case). *In the Mamba setting, the closure reduces to*

$$\left\{(X, t) \mapsto \sum_{i=1}^{d_\omega} \psi_i(\omega_t^{X,i}) + \sum_{i=1}^{d_\omega} \int_0^t \phi_i(\omega_t^{X,i} - \omega_s^{X,i}) \, d\xi_s^{X,i}\right\} \tag{70}$$

*for continuous $\psi_i : \mathbb{R} \to \mathbb{R}$ and $\phi_i : \mathbb{R} \to \mathbb{R}$.*

*Proof.* In this setting one runs in parallel $d_\omega$ diagonal systems and then takes a linear combination of the stacked hidden state. The maps in the closure of the whole system are then just the sums of maps in the closure of the subsystems. $\qquad\square$

## C   Stability and Chaining of Diagonal Systems

For this section consider, unless otherwise stated, a fixed $N \geq 0$, compact $\mathbb{X}$ and gates $(\cdot)_0, \omega, \xi$.

We will study the stability and chaining of diagonal systems defined by the choice of a matrix $V \in \mathbb{R}^{N \times d_\omega}$ such that $A_i := diag(V_i)$, where $V = [V_1 | \cdots | V_{d_\omega}]$.

Note that the present discussion holds even for non-diagonal but *commuting* matrices, since these can be simultaneously diagonalized (at the cost of considering the complex plane).

### C.1   Stability

Here we explore the stability of the dynamical system $Z^{\mathrm{X}}$, thus we need to study the eigenvalues of the $W_{s,t}^{\mathrm{X}}$. Recall how in this setting

$$W_{s,t}^{\mathrm{X}} = \exp\left(\sum_{i=1}^{d_\omega} \int_s^t A_i d\omega_r^{\mathrm{X},i}\right) = \exp\left(\mathrm{diag}(\int_s^t V d\omega_r^{\mathrm{X}})\right) = \exp\left(\mathrm{diag}\left(V(\omega_t^{\mathrm{X}} - \omega_s^{\mathrm{X}})\right)\right) \quad (71)$$

Note that because $\omega^{\mathrm{X}}$ is continuous and of bounded variation, it can be reparameterised to be Lipschitz continuous, hence absolutely continuous. Thus we can assume that $\omega^{\mathrm{X}}$ is almost everywhere differentiable and its derivative $\dot{\omega} \in L^1$.

The stability of the dynamical system then depends on the alignment between $\omega_t^{\mathrm{X}} - \omega_s^{\mathrm{X}}$ and the singular vectors of $V$. If $V\dot{\omega}_t^{\mathrm{X}} \leq 0$ for all times, where the inequality is coordinate-wise, then $W_{s,t}^{\mathrm{X}}$ has eigenvalues all in $[0, 1]$ thus the system is stable making training easier Orvieto et al. [2023a].

Consider the singular value decomposition (SVD) of the matrix $V$

$$V = \sum_{k=1}^{K} \sigma_k \, v_k u_k^\top \quad (72)$$

Then, a sufficient condition for stability is that for any $k = 1, ..., K$

$$0 > \sigma_k \in \mathbb{R}, \quad 0 \leq v_k \in \mathbb{R}^N, \quad \text{and} \quad \langle u_k, \dot{\omega}_t^{\mathrm{X}} \rangle \geq 0 \text{ for any } t \in [0, T]. \quad (73)$$

### C.1.1   The case of Mamba

In the case of Mamba Gu and Dao [2023] the matrices are diagonal and

$$d\omega_t^{\mathrm{X}} = softplus(Wx_t + \lambda)dt, \quad d\xi_t^{\mathrm{X}} = x_t \odot d\omega_t^{\mathrm{X}},$$

moreover the proposed choices of $V$ are all of type

$$V = -\mathbf{v} \otimes \mathbf{1}_{d_\omega}$$

for some choice of $0 \leq \mathbf{v} \in \mathbb{R}^N$. Note that $softmax$ is just a smooth approximation of $ReLU$ and that $Im(ReLU) \subseteq \{w \in \mathbb{R}^{d_\omega} : \langle \mathbf{1}_{d_\omega}, w \rangle \geq 0\}$ hence mamba is implicitly ensuring that the dynamical system is approximately always well-conditioned.

### C.2   Chaining

The diagonal case differs from the general one not only in the fact that the class of approximable functions is much weaker but also in the necessity for the presence of $\xi^{\mathrm{X}}$ in order to obtain any path-dependence. The term

$$\int_0^t \phi(\omega_t^{\mathrm{X}} - \omega_s^{\mathrm{X}}) \cdot d\xi_s^{\mathrm{X}}$$

becomes then a crucial component. At first sight one might think that such a term allows to recover at least level two components of the Signature of $(\omega^{\mathrm{X}}, \xi^{\mathrm{X}})$, unfortunately things are not as easy as they may seem. Notice how inside of the integral time is "going backwards" from the perspective of $\omega^{\mathrm{X}}$, thus we can in general approximate terms of type

$$\int_0^t \int_s^t d\omega_r^{\mathrm{X},i} d\xi_s^{\mathrm{X},j} = \int_{1-t}^1 \int_{1-t}^r d\overleftarrow{\omega}_r^{\mathrm{X},i} d\overleftarrow{\xi}_s^{\mathrm{X},j} = \mathrm{Sig}((\overleftarrow{\omega}^{\mathrm{X}}, \overleftarrow{\xi}^{\mathrm{X}}))_{1-t,1}^{i_\omega j_\xi}$$

which are indeed terms of the Signature, but of the reverse paths $\overleftarrow{\omega}_r^{\mathrm{X}} = \omega_{1-r}^{\mathrm{X}}$ and $\overleftarrow{\xi}_s = \xi_{1-s}^{\mathrm{X}}$!

**Proposition C.1.** *Fix a compact input set* $\mathbb{X}$*, continuous gates* $(\cdot)_0, \omega, \xi$ *and* $X_0^1 = 1$*. If the components of* $\xi^X$ *are linear combinations of those of* $\omega^X$*, with time-independent weights, then linear functionals on* $Z_t^X$ *can, uniformly in* $\mathbb{X} \times [0,1]$*, approximate arbitrarily well the following level 2 terms of* $Sig((\omega^X, \xi^X))_{0,t}$*:*

$$\int_0^t \int_0^s d\omega_r^{X,i} d\xi_s^{X,j} = Sig((\omega^X, \xi^X))_{0,t}^{i_\omega j_\xi}$$

*Proof.* Under these hypotheses we know that linear functionals on $Z_t^X$ are uniformly dense, for continuous $\psi, \phi$, in

$$\left\{ (X, t) \mapsto \psi(\omega_t^X) \cdot X_0 + \int_0^t \phi(\omega_t^X - \omega_s^X) \cdot d\xi_s^X \right\}.$$

Assume $\xi_s^{X,j} = \langle \alpha_j, \omega_t^X \rangle$ and consider the choices

$$\psi(x) = (x^i \langle \alpha_j, x \rangle, 0, \cdots, 0)^\top, \quad \phi(x) = -(0, \cdots, 0, x^i, 0, \cdots, 0). \tag{74}$$

so that

$$\psi(\omega_t^X) \cdot X_0 = \omega_t^{X,i} \xi_t^{X,j} \quad \phi(\omega_t^X - \omega_s^X) \cdot d\xi_s^X = -(\omega_t^{X,i} - \omega_s^{X,i}) d\xi_s^{X,j}. \tag{75}$$

To conclude note that

$$\omega_t^{X,i} \xi_t^{X,j} = \int_{s=0}^t \int_{r=0}^t d\omega_r^{X,i} d\xi_s^{X,j} = \int_{s=0}^t \int_{r=0}^s d\omega_r^{X,i} d\xi_s^{X,j} + \int_{s=0}^t \int_{r=s}^t d\omega_r^{X,i} d\xi_s^{X,j}$$

$$= \int_0^t \int_0^s d\omega_r^{X,i} d\xi_s^{X,j} + \int_{s=0}^t (\omega_t^{X,i} - \omega_s^{X,i}) d\xi_s^{X,j}$$

hence

$$\int_0^t \int_0^s d\omega_r^{X,i} d\xi_s^{X,j} = \omega_t^{X,i} \xi_t^{X,j} - \int_{s=0}^t (\omega_t^{X,i} - \omega_s^{X,i}) d\xi_s^{X,j} = \psi(\omega_t^X) \cdot X_0 + \int_0^t \phi(\omega_t^X - \omega_s^X) \cdot d\xi_s^X$$

$$\square$$

If $X \in C_{1,0}([0,1]; \mathbb{R}^d)$ we can use the previous result to compute its Signature entries by chaining *diagonal* Linear CDEs.

**Theorem C.2.** *Assume a compact input set* $\mathbb{X} \subset C_{1,0}([0,1]; \mathbb{R}^d)$*. For any* $I \in \mathbb{W}_d$ *with* $|I| \geq 2$ *and* $\epsilon > 0$ *there is a sequence of linear maps* $W_k \in \mathbb{R}^{N_k \times 1}$ *and weights for the following family of chained Linear CDEs*

$$dZ_t^{1,X} = \sum_{i=1}^d A_i^{(1)} Z_t^{1,X} dX_t^i + B^{(1)} dX_t \in \mathbb{R}^{N_1}, \quad Z_0^{1,X} = Z_0^1, \tag{76}$$

$$dZ_t^{k+1,X} = \sum_{i=1}^{d+1} A_i^{(k+1)} Z_t^{k+1,X} d \begin{bmatrix} W_k Z^{k,X} \\ X \end{bmatrix}_t^i + B^{(k+1)} dX_t \in \mathbb{R}^{N_{k+1}}, \quad Z_0^{k+1,X} = Z_0^{k+1}, \tag{77}$$

*such that for some* $v \in \mathbb{R}^{N_{|I|-1}}$ *one has*

$$\sup_{(X,t) \in \mathbb{X} \times [0,1]} |Sig(X)_{0,t}^I - \langle v, Z_t^{|I|-1,X} \rangle| \leq \epsilon \tag{78}$$

*Proof.* For $|I| = 2$ we can apply Prop. C.1. Assume the theorem holds for $|I| \leq k$ and let $M := \sup_{X \in \mathbb{X}} \|X\|_{1-var}$. Fix $|Ij| = k + 1$ and $W_{k-1} \in \mathbb{R}^{N_{k-1} \times 1}$ such that

$$\sup_{(X,t) \in \mathbb{X} \times [0,1]} |Sig(X)_{0,t}^I - W_{k-1} Z_t^{k-1,X}| \leq \frac{\epsilon}{M}.$$

Again by Prop. C.1 there are a $N_k$ and $v \in \mathbb{R}^{N_k}$ such that

$$\sup_{(X,t) \in \mathbb{X} \times [0,1]} |\int_0^t W_{k-1} Z_s^{k-1,X} dX_s^j - \langle v, Z_t^{k,X} \rangle| \leq \epsilon.$$

Then

$$|\text{Sig}(X)_{0,t}^{Ij} - \langle v, Z_t^{k,\text{X}} \rangle| \leq | \int_0^t \text{Sig}(X)_{0,s}^I dX_s^j - \int_0^t W_{k-1} Z_s^{k-1,\text{X}} dX_s^j | + | \int_0^t W_{k-1} Z_s^{k-1,\text{X}} dX_s^j - \langle v, Z_t^{k,\text{X}} \rangle |$$

$$\leq \int_0^t |\text{Sig}(X)_{0,s}^I - W_{k-1} Z_s^{k-1,\text{X}}||dX_s^j| + \epsilon$$

$$\leq \int_0^t \frac{\epsilon}{M}|dX_s^j| + \epsilon \leq 2\epsilon$$

thus concluding the proof. $\qquad\square$

Then Proposition 4.5 follows as a corollary by running in parallel the systems above to recover simultaneously multiple Signature entries.

### C.2.1 $ReLU$ **activation choice**

Models like *Mamba* do not only use diagonal matrices but also consider controls of a specific kind:

$$\omega_t^{\text{X}} = \int_0^t ReLU(WX_s + b)ds$$

The choice of $ReLU$ enforces $\dot\omega_t \geq 0$ for all times as seen above, but could, a priori, destroy information about $X$ which allows for the recovery, after chaining, of its Signature.

Does this choice keep some expressivity? Fortunately almost all of it: since

$$ReLU(x) - ReLU(-x) = x$$

one can choose a linear map $W$ which allows to linearly recover

$$\tilde\omega_t^{\text{X}} = \int_0^t X_s ds$$

from $\omega_t^{\text{X}}$. By correspondingly modifying the form of $\psi$ and $\phi$ in (74) such that

$$\psi(\omega_t^{\text{X}}) \cdot X_0 = \tilde\omega_t^{\text{X},i} \xi_t^{\text{X},j} \quad \phi(\omega_t^{\text{X}} - \omega_s^{\text{X}}) \cdot d\xi_s^{\text{X}} = -(\tilde\omega_t^{\text{X},i} - \tilde\omega_s^{\text{X},i})d\xi_s^{\text{X},j}. \tag{79}$$

one is able, through a similar chaining procedure, to recover arbitrarily deep entries of the Signature of $\tilde\omega_t^{\text{X}} = \int_0^t X_s ds$.

# D  Path-to-Path

**Definition D.1.** A map $G \in C^0(C_{1,0}([0,1];\mathbb{R}^d) \times [0,1];\mathbb{R})$ is *causal* iff for all $t \in [0,1]$ and paths $\omega, \tilde{\omega} \in C_{1,0}([0,1];\mathbb{R}^d)$ one has

$$\omega|_{[0,t]} = \tilde{\omega}|_{[0,t]} \implies G(\omega, t) = G(\tilde{\omega}, t)$$

*i.e.* G is *causal* if it does not look in the future.

**Proposition D.2.** *Assume a compact input set $\mathbb{X}$, continuous $(\cdot)_0, \omega, \xi$, $X_0^1 \equiv 1$ and $\omega_t^{X,1} \equiv t$. Then for all $\epsilon > 0$ and all* causal $G \in C^0(C_{1,0}([0,1];\mathbb{R}^{d_\omega}) \times [0,1];\mathbb{R})$ *there exist an integer $N \geq 0$, some Feed Forward neural network $F : \mathbb{R}^N \to \mathbb{R}$, and parameters $C \in \mathbb{R}^{N \times d_0}, A_i \in \mathbb{R}^{N \times N}, B \in \mathbb{R}^{N \times d_\xi}$ such that*

$$\sup_{X \in \mathbb{X}} \sup_{t \in [0,1]} |F(Z_t^X) - G(\omega^X, t)| < \epsilon \tag{80}$$

*Proof.* Fix $\epsilon > 0$. By B.7 the space $\mathcal{H}_{[0,1]}^{(\cdot)_0, \omega, \eta}$ contains all functionals of form

$$(X, t) \mapsto \langle \alpha, \mathrm{Sig}(\omega^X)_{0,t} \rangle$$

thus, by the properties of the signature and by compactness of $\mathbb{X}$, for any fixed $s_0 \in [0,1]$ there is some $f \in \mathcal{H}_{[0,1]}^{(\cdot)_0, \omega, \eta}$ such that

$$\sup_{X \in \mathbb{X}} |f(X, s_0) - G(\omega^X, s_0)| < \epsilon$$

Using the fact that $G \in C^0([0,1]; C^0(C_{1,0}([0,1];\mathbb{R}^{d_\omega}); \mathbb{R}))$ and compactness of $[0,1]$, we find a finite set $\{0 \leq s_0 \leq \cdots \leq s_M \leq 1\}$ of points and $f_0, \ldots, f_M \in \mathcal{H}_{[0,1]}^{(\cdot)_0, \omega, \eta}$ such that

$$\sup_{(X,s) \in \mathbb{X} \times [s_{i-1}, s_{i+1}]} |G(\omega^X, s) - G(\omega^X, s_i)| < \epsilon \tag{81}$$

$$\sup_{X \in \mathbb{X}} |f_i(X, s_i) - G(\omega^X, s_i)| < \epsilon \tag{82}$$

$$\sup_{(X,s) \in \mathbb{X} \times [s_{i-1}, s_{i+1}]} |f_i(X, s) - f_i(X, s_i)| < \epsilon \tag{83}$$

for $i = 0, \ldots, M - 1$. Notice then how for all $X \in \mathbb{X}$ and $s \in [s_{i-1}, s_{i+1}]$

$$|f_i(X, s) - G(\omega^X, s)| \leq |f_i(X, s) - f_i(X, s_i)| + |f_i(X, s_i) - G(\omega^X, s_i)| + |G(\omega^X, s_i) - G(\omega^X, s)| \leq 3\epsilon$$

It follows that the map $F \in C^0([0,1] \times \mathbb{X}; \mathbb{R})$ linearly interpolating the $f_i$ in time satisfies

$$\sup_{X \in \mathbb{X}} \sup_{t \in [0,1]} |F(X)_t - G(\omega^X, t)| < 6\epsilon$$

To conclude note that $\mathbb{X}$ being compact, the $f_i$ take values in a common compact set $K \subseteq \mathbb{R}$. There exist then a neural network $\Psi : [0,1] \times K^M \to \mathbb{R}$ such that

$$\sup_{i \in 0, \ldots, M-1} \sup_{s \in [s_i, s_{i+1}]} \left| \Psi(t, z) - \left( \frac{s_{i+1} - s}{s_{i+1} - s_i} z_i + \frac{s - s_i}{s_{i+1} - s_i} z_{i+1} \right) \right| < \epsilon$$

which means that

$$\sup_{X \in \mathbb{X}} \sup_{t \in [0,1]} |\Psi(t, f_0(X, t), \ldots, f_M(X, t)) - F(X, t)| < \epsilon$$

Recalling that $\omega_t^{X,1} = t$ we get that $X \mapsto \{t \mapsto t\} \in \mathcal{H}_{[0,1]}^{(\cdot)_0, \omega, \eta}$ so that, given density of linear maps on $Z^X$ in the space, $\Psi(t, f_0(X, t), \ldots, f_M(X, t))$ can be uniformly approximated. Triangular inequality gives finally

$$\sup_{X \in \mathbb{X}} \sup_{t \in [0,1]} |\Psi(t, f_0(X, t), \ldots, f_M(X, t)) - G(\omega^X, t)| < 7\epsilon$$

which, by arbitrariness of $\epsilon$, gives the thesis. □

The non-linearity is crucial for the path-to-path result. A map of type $(\omega, t) \mapsto \langle t\alpha, \mathrm{Sig}(\omega)_{0,t}\rangle$ cannot be approximated arbitrarily well by $(\omega, t) \mapsto \langle \beta, \mathrm{Sig}(\omega)_{0,t}\rangle$.

In any case, note that in the proof the role of the neural network is only that of interpolating the RKHS elements in the right order and at the right time. All the non-linear complexity of learning the particular $G$ is offloaded and taken care of by the RKHS elements.

*Remark* D.3. In the proof we have only considered the part of $T(X)$ concerning $\omega^{\mathrm{X}}$, but $T(X)_t$ depends linearly on $X_0$ and $\xi^{\mathrm{X}}$ suggesting that neural networks on $\mathcal{H}^{(\cdot)_0, \omega, \eta}_{[0,1]}$ have stronger generalization properties. In fact one can prove that it is possible to approximate all continuous $G(X_0, \omega^{\mathrm{X}}, \xi^{\mathrm{X}}, t)$, this is done by reconstructing $X_0$ and $\xi^{\mathrm{X}}_{[0,1]}$ as in the classical SSM case *cf.* Orvieto et al. [2023b].

# E    Wronskian Matrix Theory

In this section we obtain a unified theory studying the solutions to general Linear CDEs. The results presented here are not new and can be found in different terms in the literature Friz and Victoir [2010], despite this we have decided to reproduce them from scratch for completeness, notational reasons and to present a self-contained theory.

**Theorem E.1.** *For any choice* $\{A^1, \ldots, A^d\} \subseteq C^0([0,1]; \mathbb{R}^{N \times N})$ *and* $\omega \in C_1([0,1]; \mathbb{R}^d)$ *there exist a unique map* $W \in C^0([0,1] \times [0,1]; \mathbb{R}^{N \times N})$ *solving the following CDE*

$$W_{s,t} = Id_N + \sum_{i=1}^{d} \int_{\tau=s}^{t} A_\tau^i W_{s,\tau} d\omega_\tau^i \tag{84}$$

*Proof.* We will use Banach fixed point theorem leveraging the completeness of the space $\Omega := C^0([0,1] \times [0,1]; \mathbb{R}^{N \times N})$ with the uniform norm

$$\|X\|_\infty := \sup_{s,t \in [0,1]} \|X_{s,t}\|_{op}$$

Define the map $\Gamma : \Omega \to \Omega$ as

$$\Gamma(X)_{s,t} = Id_N + \sum_{i=1}^{d} \int_{\tau=s}^{t} A_\tau^i X_{s,\tau} d\omega_\tau^i$$

One has, for $X, Y \in \Omega$ and $k \in \mathbb{N}$ setting $\Gamma^0 = Id_\Omega$, that

$$\Gamma^{k+1}(X)_{s,t} - \Gamma^{k+1}(Y)_{s,t} = \sum_{i=1}^{d} \int_{\tau=s}^{t} A_\tau^i \left( \Gamma^k(X)_{s,\tau} - \Gamma^k(Y)_{s,\tau} \right) d\omega_\tau^i$$

which iterated gives

$$\Gamma^{k+1}(X)_{s,t} - \Gamma^{k+1}(Y)_{s,t} = \sum_{\substack{I \in \mathcal{W}_d \\ |I| = k+1}} \int_{\tau_{k+1}=s}^{t} \cdots \int_{\tau_1=s}^{\tau_2} \left( \prod_{j=k+1}^{1} A_{\tau_j}^{I_j} \right) (X_{s,\tau_1} - Z_{s,\tau_1}) \prod_{j=1}^{k+1} d\omega_{\tau_j}^{I_j}$$

$$= \sum_{\substack{I \in \mathcal{W}_d \\ |I| = k+1}} \int_{\tau \in \Delta_{[s,t]}^{k+1}} A_\tau^I (X_{s,\tau_1} - Z_{s,\tau_1}) d\omega_\tau^I$$

where $\mathcal{W}_d$ is the set of words in the alphabet $\{1, \ldots, d\}$ and

$$\Delta_{[s,t]}^k := \{(\tau_1, \ldots, \tau_k) \in [0,1]^k : \forall j \in 1, \ldots, k-1. \ \tau_j \leq \tau_{j+1}\}$$

$$A_\tau^I := \prod_{j=k+1}^{1} A_{\tau_j}^{I_j} \quad d\omega_\tau^I := \prod_{j=1}^{k+1} d\omega_{\tau_j}^{I_j}$$

By defining $M = \max\{\|A^i\|_\infty : i \in \{1, \ldots, d\}\}$ then one clearly has

$$\left\| \Gamma^k(X) - \Gamma^k(Y) \right\|_\infty \leq \frac{(dM \|\omega\|_{1-var})^k}{k!} \|X - Y\|_\infty \tag{85}$$

thus definitely (in $k$) the map $\Gamma^k$ is a contraction. By Banach fixed point there exist a unique fixed point $W \in \Omega$.    $\square$

**Theorem E.2.** *Under the assumptions of the previous theorem one can write* $W_{s,t}$ *explicitly as*

$$W_{s,t} = \sum_{I \in \mathcal{W}_d} \int_{\tau \in \Delta_{[s,t]}^{|I|}} A_\tau^I d\omega_\tau^I \tag{86}$$

*moreover if for all* $i$ *the matrix-valued maps are constant on all* $[0,1]$ *i.e.* $A_t^i \equiv A_i$ *then*

$$W_{s,t} = \sum_{I \in \mathcal{W}_d} A_I Sig(\omega)_{s,t}^I \tag{87}$$

*where* $Sig(\omega)_{s,t}^I$ *is the Signature of the path* $\omega$.

*Proof.* The second assertion follows from the first by definition of the Signature of a path.

Regarding the first notice how the series is absolutely convergent in $\mathbb{R}^{N \times N}$, uniformly in $s, t$ since

$$\sum_{I \in \mathcal{W}_d} \left\| \int_{\tau \in \Delta^{|I|}_{[s,t]}} A_\tau^I d\omega_\tau^I \right\|_{op} \leq \sum_{k=0}^{\infty} d^k M^k \frac{\|\omega\|^k_{1-var,[s,t]}}{k!}$$

$$= e^{dM\|\omega\|_{1-var,[s,t]}} \leq e^{dM\|\omega\|_{1-var,[0,1]}}$$

thus for any $s, t \in [0, 1]$ the series defines an element of $\tilde{W}_{s,t} \in \mathbb{R}^{N \times N}$.

Using the uniformity of this bound and the fact that for all $I \in \mathcal{W}_d$ one has

$$\tilde{W}^I_{s,t} := \int_{\tau \in \Delta^{|I|}_{[s,t]}} A_\tau^I d\omega_\tau^I \in \Omega$$

as a function of $(s, t)$, which moreover is uniformly continuous

$$\left\| \tilde{W}^I_{s_1,t_1} - \tilde{W}^I_{s_2,t_2} \right\|_{op} = \left\| \int_{\tau \in \Delta^{|I|}_{[s_1 \wedge s_2, t_1 \vee t_2]}} (\delta_{\tau \in \Delta^{|I|}_{[s_1,t_1]}} - \delta_{\tau \in \Delta^{|I|}_{[s_2,t_2]}}) A_\tau^I d\omega_\tau^I \right\|_{op}$$

$$\leq M^{|I|} \int_{\tau \in \Delta^{|I|}_{[s_1 \wedge s_2, t_1 \vee t_2]}} \left| \delta_{\tau \in \Delta^{|I|}_{[s_1,t_1]}} - \delta_{\tau \in \Delta^{|I|}_{[s_2,t_2]}} \right| |d\omega_\tau^I|$$

$$\leq M^{|I|} \|\omega\|^{|I|}_{[s_1 \wedge s_2, s_1 \vee s_2] \cup [t_1 \wedge t_2, t_1 \vee t_2]},$$

one concludes that $\tilde{W}_{s,t} \in \Omega$. Finally notice that $\tilde{W}_{s,t}$ is a fixed point of $\Gamma$

$$\Gamma(\tilde{W}_{s,t}) = Id_N + \sum_{i=1}^{d} \int_{\tau=s}^{t} A_\tau^i \left( \sum_{I \in \mathcal{W}_d} \int_{\tau \in \Delta^{|I|}_{[0,1]}} A_\tau^I d\omega_\tau^I \right) d\omega_\tau^i$$

$$= Id_N + \sum_{\substack{I \in \mathcal{W}_d \\ |I| \geq 1}} \int_{\tau \in \Delta^{|I|}_{[0,1]}} A_\tau^I d\omega_\tau^I d\omega_\tau^i$$

$$= \sum_{I \in \mathcal{W}_d} \int_{\tau \in \Delta^{|I|}_{[0,1]}} A_\tau^I d\omega_\tau^I = \tilde{W}_{s,t}$$

and conclude by uniqueness. $\qquad\square$

**Proposition E.3.** *Under the previous conditions, the unique solution of the $N$-dimensional CDE*

$$dX_t = X_0 + \sum_{i=1}^{d} \int_{\tau=0}^{t} A_\tau^i X_\tau d\omega_\tau^i \tag{88}$$

*is given by*

$$X_t = W_{0,t} X_0 \tag{89}$$

*Proof.* The solutions are unique by standard results Friz and Victoir [2010][Thm. 3.7], moreover

$$W_{0,t} X_0 = \left( Id_N + \sum_{i=1}^{d} \int_{\tau=0}^{t} A_\tau^i W_{0,\tau} d\omega_\tau^i \right) X_0 = X_0 + \sum_{i=1}^{d} \int_{\tau=0}^{t} A_\tau^i (W_{0,\tau} X_0) d\omega_\tau^i$$

$\qquad\square$

**Proposition E.4.** *The Wronskian matrix has the following properties:*

1. $\forall r, s, t \in [0, 1]. \quad W_{r,t} = W_{s,t} W_{r,s}$

2. $\forall s, t \in [0, 1]. \quad W_{s,t}^{-1} = W_{t,s}$

3. $\forall s, t \in [0, 1]. \quad W_{s,t} = Id_N + \sum_{i=1}^{d} \int_{\sigma=s}^{t} W_{\sigma,t} A_\sigma^i d\omega_\sigma^i$

*Proof.* Regarding the first statement notice that for all $X_0 \in \mathbb{R}^N$ one has

$$\tilde{X}_t := W_{s,t} W_{r,s} X_0 = \left( Id_N + \sum_{i=1}^{d} \int_{\tau=s}^{t} A_\tau^i W_{s,\tau} d\omega_\tau^i \right) W_{r,s} X_0$$

$$= W_{r,s} X_0 + \sum_{i=1}^{d} \int_{\tau=s}^{t} A_\tau^i (W_{s,\tau} W_{r,s} X_0) d\omega_\tau^i$$

$$= W_{r,s} X_0 + \sum_{i=1}^{d} \int_{\tau=s}^{t} A_\tau^i \tilde{X}_\tau d\omega_\tau^i$$

and by the previous proposition also

$$X_t := W_{r,t} X_0 = W_{r,t} X_0 + \sum_{i=1}^{d} \int_{\tau=r}^{t} A_\tau^i X_\tau d\omega_\tau^i$$

thus $X_t$ and $\tilde{X}_t$ solve the same $CDE$ and coincide at time $t = s$. This means, by uniqueness, that $X_t$ and $\tilde{X}_t$ coincide for all times; hence $W_{s,t} W_{r,s}$ and $W_{r,t}$ coincide for all times too since for any choice of $X_0$ one has $W_{s,t} W_{r,s} X_0 = W_{r,t} X_0$.

The second statement follows from the previous one setting first $r = t$ and subsequently exchanging $s$ and $t$.

To prove the third equality note that

$$0 = d_s(W_{s,t} W_{t,s}) = (d_s W_{s,t}) W_{t,s} + W_{s,t}(d_s W_{t,s})$$

hence

$$d_s W_{s,t} = -W_{s,t}(d_s W_{t,s}) W_{t,s}^{-1} = -W_{s,t} \left( \sum_{i=1}^{d} A_s^i W_{t,s} d\omega_s^i \right) W_{t,s}^{-1} = -\sum_{i=1}^{d} W_{s,t} A_s^i d\omega_s^i$$

$\square$

**Proposition E.5** (Liouville's Formula)**.** *Under the assumptions of the previous theorems, if $\omega \in C^1([0,1]; \mathbb{R}^d)$ then*

$$det(W_{s,t}) = 1 + \sum_{i=1}^{d} \int_{\tau=s}^{t} tr(A_\tau^i) det(W_{s,t}) d\omega_\tau^i = \exp\left( \sum_{i=1}^{d} \int_{\tau=s}^{t} tr(A_\tau^i) d\omega_\tau^i \right) \quad (90)$$

*Proof.* This just follows from the classical case since we can write

$$\sum_{i=1}^{d} \int_{\tau=s}^{t} A_\tau^i W_{s,\tau} d\omega_\tau^i = \int_{\tau=s}^{t} \left( \sum_{i=1}^{d} A_\tau^i \dot{\omega}_\tau^i \right) W_{s,\tau} d\tau$$

$\square$

We can now state the main result of the section:

**Theorem E.6.** *Under the assumptions of the previous theorems, given continuous functions $\{B^1, \ldots, B^t\} \in (\mathbb{R}^d)^{[0,1]}$ the unique solution of the $N$-dimensional CDE*

$$X_t = X_0 + \sum_{i=1}^{d} \int_{\tau=0}^{t} \left( A_\tau^i X_\tau + B_\tau^i \right) d\omega_\tau^i \quad (91)$$

*is given* explicitly *by*

$$X_t = W_{0,t} X_0 + \sum_{i=1}^{d} \int_{0}^{t} W_{s,t} B_s^i d\omega_s^i \quad (92)$$

*where $W_{s,t} \in C^0([0,1] \times [0,1]; \mathbb{R}^{N \times N})$ is the Wronskian matrix defined by*

$$W_{s,t} = \sum_{I \in \mathcal{W}_d} \int_{\tau \in \Delta_{[s,t]}^{|I|}} A_\tau^I d\omega_\tau^I \quad (93)$$

*Proof.* Given the unique solution $X_t$ one has

$$
\begin{aligned}
d_s(W_{s,t}X_s) &= d_s(W_{s,t})X_s + W_{s,t}d_s(X_s) \\
&= \sum_{i=1}^{d} \left( -W_{s,t}A_s^i X_s + W_{s,t}A_s^i X_s + W_{s,t}B_s^i \right) d\omega_s^i \\
&= \sum_{i=1}^{d} W_{s,t}B_s^i d\omega_s^i
\end{aligned}
$$

hence

$$
X_t - W_{0,t}X_0 = W_{t,t}X_t - W_{0,t}X_0 = \sum_{i=1}^{d} \int_{s=0}^{t} W_{s,t}B_s^i d\omega_s^i
$$

$\square$

# F ZOH and Exact Solutions

Consider a Linear CDE as the one of (25)

$$dZ_t = \sum_{i=1}^{d_\omega} A_i Z_t d\omega_t^i + B d\xi_t$$

and recall how the solution can be explicitly written, for times $s < t$, as

$$Z_t = W_{s,t} Z_s + \int_s^t W_{r,t} B d\xi_r$$

Assume moreover that in the interval $[s, t]$ both drivers have constant derivative *i.e.*

$$\omega_r = \omega_s + \boldsymbol{w}(r - s) \quad \xi_r = \xi_s + \boldsymbol{v}(r - s)$$

Then if $\mathbb{A}_{\boldsymbol{w}} := \sum_{i=1}^{d_\omega} A_i \boldsymbol{w}^i$ we get that $W_{r,t} = e^{\mathbb{A}_{\boldsymbol{w}}(t-r)}$ thus

$$Z_t = e^{\mathbb{A}_{\boldsymbol{w}}(t-s)} Z_s + \int_s^t e^{\mathbb{A}_{\boldsymbol{w}}(t-r)} B \boldsymbol{v} dr = e^{\mathbb{A}_{\boldsymbol{w}}(t-s)} Z_s + \left( \int_s^t e^{\mathbb{A}_{\boldsymbol{w}}(t-r)} dr \right) B \boldsymbol{v} \qquad (94)$$

But the integral can be explicitly solved as

$$\int_s^t e^{\mathbb{A}_{\boldsymbol{w}}(t-r)} dr = \left( -\mathbb{A}_{\boldsymbol{w}}^{-1} e^{\mathbb{A}_{\boldsymbol{w}}(t-r)} \Big|_{r=s}^t \right) = \mathbb{A}_{\boldsymbol{w}}^{-1} \left( e^{\mathbb{A}_{\boldsymbol{w}}(t-s)} - \mathbb{I} \right) \qquad (95)$$

leaving us with

$$Z_t = e^{\mathbb{A}_{\boldsymbol{w}}(t-s)} Z_s + \mathbb{A}_{\boldsymbol{w}}^{-1} \left( e^{\mathbb{A}_{\boldsymbol{w}}(t-s)} - \mathbb{I} \right) B \boldsymbol{v} \qquad (96)$$

which, setting $\Delta = t - s$, can be rewritten as

$$Z_t = e^{\mathbb{A}_{\boldsymbol{w}} \Delta} Z_s + (\mathbb{A}_{\boldsymbol{w}} \Delta)^{-1} \left( e^{\mathbb{A}_{\boldsymbol{w}} \Delta} - \mathbb{I} \right) (B\Delta) \boldsymbol{v} \qquad (97)$$

*i.e.* exactly the ZOH scheme.

