# OpenReview forum: "Theoretical Foundations of Deep Selective State-Space Models"
_NeurIPS.cc/2024/Conference — NeurIPS 2024 poster_

### Official Review · Reviewer_1RLC · 2024-07-06

**Soundness:** 3
**Presentation:** 3
**Contribution:** 3
**Rating:** 5
**Confidence:** 4

**Summary:**

This work establishes density results of input-controlled differential equations, which formulate different types of state-space models such as S4 and Mamba in the continuous-time idealization. Under this framework, different closures of these models are derived, indicating their distinct inductive biases in the (universal) approximation sense, particularly implying the advantage of data-dependent modeling.

**Strengths:**

1. This work provides fundamental theoretical justifications on the expressivity of different types of SSMs.
2. The derived inductive biases clearly distinguish the recent new data-dependent modeling (e.g. Mamba) from the classic original data-independent modeling (e.g. S4) thorough a input-selective framework, which shows rigorously the superiority of the former architecture.
3. Other theoretical results also lead to useful insights. For instance, diagonal recurrence weakens the approximation capability, but this can be alleviated via stacking.
4. All these insights are basically verified by numerical experiments.

**Weaknesses:**

1. In theory, only the density type of results is provided. That is, although models can rates certain targets universally when models' parameters go to infinity, the convergence *rates* are not characterized. This is much more important since the convergence can  be rather slow in certain situations (e.g. curse of dimension), leading to possibly vacuous bounds. The theoretical part would be strengthened and more convincing if authors can discuss more on the approximation rates of SSMs, especially the improvements (in parameter efficiency) of data-dependent modeling.
2. In experiments, the simulations are conducted on low-dimensional (2 & 3) and synthetic datasets. Is it due to the ability of path signature? Can the path signature be powerful and efficient when handling high-dimensional (and real-world) input data?

**Questions:**

1. Please provide more details of questions raised in the weaknesses section above.
2. In formulation, the mentioned "gating" mechanism in this work seems to have a different meaning as usual. Here, gating refers to the transformation of inputs (i.e. data pre-processing). However, in practice, gating often means the (multiplicative) interaction between the hidden states and readout layers (i.e., to get outputs). Can authors explain more about this?
3. Minor issues: What is the definition of (linear) NCDE (see line 311, 312, 315, 317 and 643).

**Limitations:**

As is stated by authors, this work applies a continuous-time idealization in the formulation. The role of concrete discretization schemes is also worthy of explorations, particularly under the case of data-dependent discretization.

---

> ### Author Rebuttal · Authors · 2024-08-07
>
> We thank the reviewer for their thoughtful feedback on our paper. We appreciate the positive assessment of our work's soundness, presentation, and contribution. Below, we address the raised points.
>
> ## Weaknesses
>
> - **Rates of convergence**: While our current theoretical framework in general provides only density-type results, we agree that the characterization of convergence rates is a crucial question. We envisioned this paper as one establishing the language and theoretical groundwork for further analysis in such a direction.
> We want to point out that for Theorem 4.2, the bounds are explicit in $N$, as given in equation (55).
> Nevertheless, a study of the rates in the general case, particularly for the stacked diagonal systems, would be more technically involved and is considered beyond the scope of the current work. However, we acknowledge its importance and will highlight it as a critical direction for future research.
>
> - **Experiments**: We would like to clarify a potential misunderstanding: our approach does not utilize the path signature as part of the method itself. Instead, we are simply using specific terms in the path signature as the labels for the synthetic datasets.
> We opted for low-dimensional, synthetic datasets to provide empirical evidence for the theoretical results in our paper in a simple setting. These experiments were not intended to comment on the effectiveness of the path signature when handling high-dimensional data. There are a number of interesting works which apply variants of the signature to real-world data, with some recent examples being the signature kernel (https://arxiv.org/abs/2006.14794, https://arxiv.org/abs/2006.05805), the path development layer (https://arxiv.org/abs/2204.00740), the randomized signature (https://arxiv.org/abs/2201.00384), and log neural controlled differential equations (https://arxiv.org/abs/2402.18512).
>
> ## Questions
>
> - See above
> - **Gating**: Our definition of gating is aligned to that of forget and input gate in the LSTM, GRU and SSM literature. In particular, what the reviewer refers to is known as output gate: what instead $\omega$ and $\xi$ modulate is how inputs are selected and forgotten. The multiplicative interaction crucial in this setting is $Z_t^X d\omega_t^X$ - this can be thought of as a forget gate on the hidden state, with input-dependent gating function induced by $\omega$. We note that the gate terminology was introduced in the LSTM paper, but is also used in recent SSMs paper such as in Mamba and Griffin.
>
> - **NCDE**: NCDE stands for Neural CDE. In fact a linear NCDE is just a linear CDE, hence there was no need to mention the “Neural” part – we used this terminology since it is linked to some results on rough path theory applied to neural networks (e.g. https://arxiv.org/abs/2005.08926) but here this connection is not necessary. We thank the reviewer for pointing this out. We will revise the manuscript to use the term “CDE” instead of “NCDE” in the relevant sections to avoid any confusion.
>
> ## Limitations
>
> We agree with the reviewer on this point. We believe that our casting of SSMs in the language of Rough Paths theory might open the way for such a study of discretizations, using the honed tools of the field.

---

> > ### Comment · Reviewer_1RLC · 2024-08-12
> >
> > Thanks for the reply. I still feel that the convergence rate characterization is much more important than the density analysis. Certainly, eq. (55) derives the dependence on hidden dimensions, but for sequence modeling problems, the approximation rate regarding the *input dimensions* and *dynamical properties* is more crucial, and the numerical verification here is performed only for low-dimensional tasks. Does the current analysis framework have the potential to solve this difficulty?

---

> > > ### Author Response · Authors · 2024-08-13
> > >
> > > We thank the reviewer for the message and the additional questions.
> > >
> > > We agree that including insights on the effects of width is interesting; see next paragraph. However, note that width alone is insufficient to capture all the high-order  terms of the signature in the diagonal (Mamba-like) case. Our objective in this paper, besides completely characterizing the density, is to highlight a fundamental limitation of diagonal models that is independent of width (this is not an upper bound - our result is tight). Our insights can guide further analysis, such as the potential design of minimal epsilon-sparse recurrences that still capture the full effects of the signature without severely affecting speed or parameter count.
> > >
> > > Regarding your question: It is possible to derive width-dependent bounds, yet these are unlikely to be tight, similar to what is observed in standard MLPs.
> > >
> > > We will develop on this in a new subsection in our revision. Techniques and results going in this direction are actively researched by the Rough paths in ML community. As a primal example we would like to refer you to the paper titled "Generalization Bounds for Neural Controlled Differential Equations" (https://arxiv.org/abs/2305.16791). This work provides a generalization bound for a broader class of learners, specifically Neural Controlled Differential Equations (NCDEs), which, as explained in our previous response, are closely related to Linear CDEs, offering a detailed analysis of both generalization bounds and approximation biases. In order to do this the data streams and models are studied from a rough path perspective, dynamical bounds are produced by considering Lipschitz properties of the vector fields and the regularity of the input streams via their 1-variation. These are the natural tools to employ with CDEs like the ones we specify in this work.
> > >
> > > What we just described is an example of how our work establishes a solid theoretical connection between State Space Models (SSMs) and Rough Paths, setting the stage for further studies, and allowing researchers to leverage the rich literature of existing results.

---

### Official Review · Reviewer_xt92 · 2024-07-13

**Soundness:** 3
**Presentation:** 2
**Contribution:** 3
**Rating:** 6
**Confidence:** 2

**Summary:**

This paper proposes a framework for better understanding key features that allow the success of SSMs. To be specific, the authors first show that recent SSM-based models are linear controlled differential equations (CDEs). Then, the expressive power of linear CDEs are explored, depending on whether the matrices $A_i$ are diagonal.

**Strengths:**

* This paper focuses on timely and important research problem.
* I could not fully read the proof, but the results seem correct.

**Weaknesses:**

* It would be better to provide preliminaries on rough path theory for readers who are not familiar with that.
* Although the mathematical results are interesting, I am not sure about the implications of the theoretical results in practice. It would be better if the empirical results are designed to provide the messages that are useful in practice.

**Questions:**

* How can we derive Eq.3? Why suddenly exponential term appears? Any reference for the zero-order hold discretization? (Actually, I searched for such terminology, but could not make a connection between it and Eq.3)

**Limitations:**

* Check the weakness part & questions part

---

> ### Author Rebuttal · Authors · 2024-08-07
>
> We thank the reviewer for their thoughtful feedback on our paper. We appreciate the positive remarks regarding the timeliness and importance of the research problem we address. We would like to address the points raised concerning weaknesses and questions.
>
>
> ## Weaknesses
>
> - **Preliminaries on Rough Path Theory**: We understand the importance of making our paper accessible to readers unfamiliar with Rough Path Theory. In Appendix A (referenced at line 268), we provide an introduction to the Signature Transform and reference several in-depth studies and presentations of Rough Path Theory in the context of machine learning. Additionally, in Appendix E, we provide a self-contained theory studying the solutions to general Linear CDEs. To enhance clarity and ease of access, we will ensure to reference both Appendix A and Appendix E more prominently in the main body of the paper.
>
>  - **Implications in Practice**: Our results fit into an ever-growing theoretical literature on the implicit limitations deriving from architectural choices in the context of SSMs (see e.g. https://arxiv.org/pdf/2404.08819, https://arxiv.org/abs/2406.05045), which we enhance by providing explicit, analytical characterizations and by establishing a theoretical foundation, in terms of Rough Path theory, which we envision as a base for further study. The take-away from this literature is clear: devising a non-diagonal, input-dependent, and efficiently computable transition mechanism would allow you to overcome the expressive limitations of present SSMs without the need for an arbitrarily high number of layers. A promising avenue is using low-rank or even highly sparse weights, as suggested by recent works (e.g. https://arxiv.org/abs/2310.16597, https://arxiv.org/abs/2407.08459, https://arxiv.org/abs/2406.05045).
>
> - **Further experiments**: As a further example of the practical implications of our theoretical results, we have performed additional experiments focused on the A5  benchmark introduced by https://arxiv.org/pdf/2404.08819. This benchmark is designed to evaluate models on their state-tracking, a crucial ability for solving problems that involve permutation composition, such as tracking chess moves. A key result of their paper is that state-space models such as Mamba require stacking in order to perform well on this benchmark. Our experiments have shown that even on the longest sequence length in the benchmark, where Mamba requires 4 stacked layers to achieve >90% test accuracy, a linear CDE with a trainable transition matrix requires only one layer to achieve >90% test accuracy. We intend to include a full discussion of these results in the final version of our paper.
>
>
> ## Questions
>
> - **Derivation of Eq.3 and the Appearance of the Exponential Term**: We appreciate the reviewer's attention to the details of our derivation. A self-contained explanation of Zero-Order Hold (ZOH) discretization is provided in Appendix F (referenced just before 164). To increase clarity, we will highlight the reference to this section in the main body of the paper. We also note that ZOH discretization is a conventional nomenclature for such a scheme in the SSM literature (see e.g. https://arxiv.org/abs/2303.06349).
>
> We hope these clarifications address the reviewer's concerns and enhance the overall readability and impact of our paper.

---

### Official Review · Reviewer_K8Nc · 2024-07-14

**Soundness:** 3
**Presentation:** 3
**Contribution:** 4
**Rating:** 7
**Confidence:** 5

**Summary:**

This paper proposes a framework of using Rough Path Theory to understand the expressivity of SSMs and Mamba. The paper establishes connections to linear CDEs and then uses tools from Rough Path Theory to explain why gates are so powerful in SSM models.

**Strengths:**

This is a really nice theory explaining SSMs. It gives a new way of looking at the expressivity/quality results that have been observed in prior work. Hopefully this theory can lead to new insights about how to design better SSMs in the future.

Really solid work.

**Weaknesses:**

Not much weaknesses in the work itself. The ultimate test of theory is its predictive power - the paper would be stronger if the theory could "close the loop" and propose a modification to SSM layers that would enable further performance. However, this is a high bar to clear, and I think the work stands on its own as a solid contribution even without a methodological contribution.

**Questions:**

Can you think of any improvements to SSM layers/models that the RPT theory would suggest?

---

> ### Author Rebuttal · Authors · 2024-08-07
>
> We thank the reviewer for their thoughtful feedback on our paper. We appreciate the positive remarks on the theory's presentation and contribution. We would like to address the points raised regarding “closing the loop”.
>
> Our paper shows how non-diagonal transitions lead to a substantial increase in expressivity. However, this theoretical insight is currently impeded by the reality of computation with dense layers, which is practically infeasible due to the associated computational costs.
> At the same time, recent literature (e.g. https://arxiv.org/pdf/2402.19427, section 4.2) has shed light on how the computation of linear diagonal RNNs is dominated by memory transfers. This indicates that there is leeway allowing for an increase in the complexity of the sequential mechanism, presenting itself as a very promising area of research.
>
> Such a mechanism would have to be non-diagonal but efficiently computable. Recent works (e.g. https://arxiv.org/abs/2310.16597, https://arxiv.org/abs/2407.08459, https://arxiv.org/abs/2406.05045) suggest that low-rank or even highly sparse weights should lead to the same limiting behaviors found in the dense case. This is a promising avenue, which we have not studied in this work, as it would require substantial theoretical justification as well as thorough empirical evaluations at scale. As the purpose of this paper is to outline the hypothesis class and power of SSM variants, we decided to leave this avenue for future research.
>
> However, given the importance and interest of such observations, we will augment the concluding section with them in the camera ready version, using the additional available space.

---

> > ### Comment · Reviewer_K8Nc · 2024-08-13
> > **Response**
> >
> > Thank you for the rebuttal. I will be keeping my high score.

---

### Official Review · Reviewer_KnKG · 2024-07-15

**Soundness:** 3
**Presentation:** 3
**Contribution:** 3
**Rating:** 7
**Confidence:** 2

**Summary:**

This work analyses the modeling capability of different SSMs (S4-S6 and others) using Rough Path Theory by viewing SSMs as (input-) controlled differential equations (CDE). To this end, the authors show that SSMs with with dense transition matrices (A) are able to approximate arbitrarily close any continuous functions of the inputs. In contrast, SSMs with diagonal transition matrices lack this property. However, when stacking/chaining multiple diagonal CDEs (as is the case for the deep architectures), full expressivity is gained again as the depth approaches infinity. The work is mostly theoretical, and, hence, there is only a single toy experiment showing RMSE for Mamba and S5 models with depth 1 and 2, showing that the more limited models are not able to solve the toy task, as they cannot approximate the corresponding function.

**Strengths:**

- The paper is generally well written and the authors managed to make the very involved theory and proofs more accessible by walking through the interpretation and takeaways on a more high level.
- The theoretical results are important contributions for categorizing the expressivity of existing SSM models and may potentially inform about future architecture design of SSMs and related models.

**Weaknesses:**

Despite the well-described theoretical contributions and insights, it is still not very clear to me what to take away practically.
In particular, it is not clear how useful the result regarding chaining diagonal CDEs is in practice: while it is important to have a proof that an infinite number of chains of diagonal CDE with mixing can recover the full expressiveness of the dense CDEs, the infinite case is not practically important. Instead, it would be important for practical considerations (computational efficiency vs model expressiveness) to know whether one should use dense CDEs or diagonal CDEs with mixing to achieve a certain level of expressivity.  However, as far as I understand, there is no result that quantifies e.g. how many diagonal plus mixing layers have a similar expressiveness as a certain number of dense CDEs. The authors speculate in the summary that architectures with non-diagonality might improve performance, but I do not see how this actually follows from this work, since there is no comparison between diagonal+mixing and dense in the regime of finite number of parameters.

The experimental section is very weak. While this work is mostly theoretical, this work would nevertheless be improved a lot by a greater experimental section that validates the claims. Showing error curves with a single random seed is not that interesting. I would be better to have a table of final performance that includes error bars. Furthermore, it would be great to explore a few different tasks and several ablations such as number of layers. This would make it at least empirically possible to compare dense and diagonal + mixing and random dense + learned C (or MLP) in the finite parameter setting, where its not clear what to take away from the theorems.

**Questions:**

- what is NCDE? This was not introduced.
- does random linear CDE perform best for Area computation and is also a strong baseline for Volume?  I understand that with high probability we can just take a random initialized A matrix and train only the C matrix (Theorem 4.2) to achieve full expressivity. However, that model should still not be better than a dense CDE where everything is trained such as in case of S5. Or did you choose S5 with diagonal transition?

**Limitations:**

Just a single limitation is mentioned in the conclusion. I am sure, the authors can think of many more limitations, such as my main concern raised above regarding the comparison of different mixed diagonal + mixing vs dense architectures. Or the limited experimental evidence.

---

> ### Author Rebuttal · Authors · 2024-08-07
>
> We thank the reviewer for their constructive feedback on our theoretical work. We are pleased you found our expressivity results important and our proofs accessible. We spent a considerable amount of time making our manuscript easy to parse despite the high technicality of the content.
>
> ## Weaknesses
>
> The reviewer is right in saying that our work does not explore the finite-regime setting and does not compare architectural options in terms of model parameters needed to achieve a desired level of expressivity. This would be highly desirable but is not an easy task; in addition results can be confounded by issues such as optimization, inductive biases (OOD generalization), GPU bottlenecks (e.g. memory dependency on sequence length) etc. Indeed, current research did not yet unveil such clean comparisons even for Transformers vs SSMs : the attention mechanism has more parameters compared to the S6 block in Mamba, but how exactly expressivity compares at a fixed compute budget or model dimension is unclear.
>
> While touching on the issues above is for sure needed when proposing a new model, we like to point out that the purpose of our work is not suggesting a new mixing component (i.e. dense RNNs linear in the state), but instead exploring the source of limitations of modern sequence mixing components at generic widths. To do so, we study dense linear (in the state) recurrences, and then dive into the diagonal setting. As such, our dense linear CDE framework is not intended for direct implementation in a deep model, but provides an upper bound on the expressivity which is achievable with one RNN (SSM) linear in the state but nonlinear in the input.
>
> Note that
> - **What happens at finite width**. Our results prove that one linear CDE layer can approximate any nonlinear transformation, while diagonal CDEs (e.g. Mamba) cannot. This is however only a subset of our results. Leaving out chaining from the discussion here, our theorems identify the functional form for dense and diagonal settings in closed form – this holds at any width. Specifically, why the output in the dense setting can be written as $\int_0^t \Phi(\omega^{X}_{[s,t]}) \cdot d\xi^{X}_s$ (eq 7), for width-dependent function $\Phi$ determined by the model parameters. Instead, in the diagonal setting, the functional form for the output is restricted to $\int_0^t \phi(\omega^{X}_t - \omega^{X}_s) \cdot d\xi^{X}_s$. Note that width can only modify the complexity of the nonlinear functions $\Phi$ and $\phi$, but cannot modify the output structure.
>
> - **Experiments**. While we believe very much that experiments can provide valuable insights in some setting, our objective here is to provide a theoretical foundation, hence the title. We remark that our results are not bound - they provide a tight description of the fundamental elements interacting when outputs are constructed, with model dimension only controlling functions $\Phi$ and $\phi$ above. We also note that our results are novel - also in the realm of Rough Paths theory as an independent mathematical subject. We refer the reviewer to the supplementary PDF for augmented and new results, as discussed in the **general comment**.
>
>
> ## Questions:
>
> - **NCDE**: NCDE stands for Neural CDE. In fact a linear NCDE is just a linear CDE, hence there was no need to mention the “Neural” part – we used this terminology since it is linked to some results on rough path theory applied to neural networks (e.g. https://arxiv.org/abs/2005.08926) but here this connection is not necessary. We thank the reviewer for pointing this out. We will revise the manuscript to use the term “CDE” instead of “NCDE” in the relevant sections to avoid any confusion.
>
> - **S5**. As you stated, we expect that a dense CDE with a trainable transition matrix would be the best performing model on this benchmark. However, the computational burden of training such a model is very high. In practice, S5 is parameterized using a diagonal transition matrix, as discussed in Section 3.2 of the S5 paper (https://arxiv.org/abs/2208.04933). Consequently, the same theoretical results on expressivity apply to both S4 and S5. In fact, any model where $d\omega$ is $1$-dimensional (i.e. any model with only one transition matrix $A_1$) is equivalent over $\mathbb{C}$ to a diagonal model, by virtue of $A_1$ being similar to a diagonal matrix. To obtain the full expressivity results more than one $A_i$ must be present, and these matrices must not be simultaneously diagonalizable. As demonstrated in the proof of Theorem 4.3, in the diagonal case, the relevant terms of the signature are those appearing in its symmetric part. The signature of a $1$-dimensional path being fully symmetric shows consistency of our findings. On reflection, given this equivalence, we believe introducing S5 in our experiments is redundant. To streamline our presentation and strengthen the connection between our theoretical and experimental results, we have chosen to replace S5 with S4 in our experiments. We hope this revision addresses your concerns.
>
>
> ## Limitations:
>
> We will augment the list of limitations in the camera ready version, using the additional available space.

---

> > ### Comment · Reviewer_KnKG · 2024-08-11
> > **Response to rebuttal**
> >
> > Thank you for your response.
> >
> > Note that I did not question the novelty of this work and also agree that the theoretical results are a great contribution, which is why I voted to accept the paper. However, I also remain of the opinion that the theoretical results do not offer a direct and very practical suggestion regarding design of new architectures. This is, because the proofs are indeed *not* independent of the width if you consider practical settings with multiple layers (chaining) and non-linearities in between. I do not want to suggest that such a proof should have been made, because it very likely is not possible. However, it remains not clear from this work, e.g. how much chaining would be sufficient with diagonal matrices and whether this would actually be a problem for Mamba, given that it a very deep model.
> >
> > I also understand that the contributions of this work are of theoretical nature and do not expect a very experimental section. However, I do still think that a bit more than Fig. 1 to support the theoretical findings would have been not only insightful, but important to underscore these findings.

---

> > > ### Author Response · Authors · 2024-08-13
> > >
> > > We appreciate your continued feedback and your recognition of the theoretical contributions of our work. We understand your concerns regarding the practical implications of our results and the empirical validation to support our theoretical findings.
> > >
> > > **Regarding your point about the practical impact of our work**:
> > >
> > > We agree that understanding the implications of chaining diagonal matrices and the expressivity of models like Mamba in practical scenarios with multiple layers is important. While our work does not offer a direct blueprint for designing new architectures, we believe it lays the groundwork for future research in this area by identifying key components that could influence expressivity. In particular the fact that diagonal systems capture only the symmetric part of the signature could be leveraged to extend the chaining results to the non-linear case: approximating the result with a linear functional on the symmetric part of the signature, then leveraging the algebra structure of the shuffle product. This procedure would show how even $n$ chained diagonal (Mamba-like) layers, interleaved with non-linearities, would capture the non-symmetric part of the signature fundamentally only up to level $n$. Note that width alone cannot capture the higher terms of the signature; what we point out is a fundamental and *width-independent* drawback of diagonal models, a limitation which can serve as a guide for further avenues of analysis. We also note that it can be possible to derive width-dependent bounds, however those would likely not be tight (as in the standard MLP case).
> > >
> > > **Regarding the empirical validation**:
> > >
> > > We understand the importance of supporting theoretical claims with empirical evidence. As mentioned in our initial response, we have indeed conducted new experiments, including a novel analysis using the A5 benchmark of Merril et al. 2024 (“Illusion of state” paper). These experiments were designed to provide additional insights into the practical performance of different architectures, particularly focusing on the depth required for models like Mamba to achieve high performance on state tracking (a task that cannot be easily solved by attention). We included these results in the supplementary material and in the general response pdf.
> > >
> > > Given your feedback, it seems that these new experiments may not have been fully considered. We are sorry if we perhaps did not emphasize them. If this is the case, we encourage you to review the supplementary material pdf and general response where these additional results are discussed in detail.
> > >
> > > We value your constructive feedback and would be open to any further suggestions on how we could better present or highlight these results in the final version of our paper.

---

### Author Rebuttal · Authors · 2024-08-07

We would like to extend our gratitude to all reviewers for their insightful comments and valuable feedback. We appreciate the time and effort invested in evaluating our work. Below, we address the primary clarifications about relevance and practical implications.

- **Our Contribution**. Ours is a theoretical paper studying approximation of sequence to sequence maps with modern (gated) state-space models. Our results are to be inserted in the vast literature on expressivity and computational power of recurrent mechanisms, but with a fresh look at modern architectural components (e.g. the Mamba block). Our results are novel, and our tools draw a strong connection to powerful techniques in Rough Path Theory. We provide a closed-form analytical characterization of the class of learnable functions implemented by SSM variants, and discuss how this is affected by computationally critical choices such as the use of diagonal matrices. Our results on universality hold in the width limit, as common in much of the deep learning literature, but crucially the input mixing mechanics we identify provide valuable insights also at finite width, acting as upper limits to computational power.

While our paper is mainly intended for readers interested in deepening our theoretical understanding of new deep learning blocks, we are also concerned with practical implications and the road ahead for future research:

- **Relevance of the results**. Most importantly, the functional forms we identify using Rough Path Theory reveal how input tokens are processed in dense (idealized) versus diagonal (Mamba-like) models. In diagonal models only first order information about the input sequence is used when producing an output, whereas, in the dense setting the entire history contributes to the computation. This is a significant difference, independent of width. Recent investigations, such as those found in https://arxiv.org/abs/2209.11895, https://arxiv.org/pdf/2402.01032, https://arxiv.org/pdf/2404.08819, and https://arxiv.org/abs/2312.04927, have explored similar distinctions in token processing strategies. Compared to these works, our results also offer tight guarantees on expressive power and on the effects of chaining.

- **New Experiments**.  We have rerun the signature prediction experiment for each model with 5 different random seeds. We have augmented our plot of the validation RMSE to show the range of the validation RMSE over the 5 runs, and this can be found in the supplementary PDF file. We have also conducted additional experiments on the A5 benchmark introduced by https://arxiv.org/pdf/2404.08819. This benchmark is designed to evaluate models on their state-tracking ability, which is crucial for tasks involving permutation composition, such as tracking chess moves. A key finding from their paper is that state-space models like Mamba require stacking to perform well on this benchmark. Our experiments demonstrate that even for the longest sequence length in the benchmark, where Mamba needs 4 stacked layers to achieve over 90% test accuracy, a linear CDE with a trainable transition matrix achieves over 90% test accuracy with only one layer. We have included plots of these new results in the supplementary PDF file and we plan to include a broader discussion of these results in the final version of our paper.

- **Practical Implications**. Our results outline some inherent limitations of diagonal recurrent computation. We are not making claims of the type "diagonal is less expressive, use dense”; even though dense recurrences are more expensive. However, implications and directions for future research are clear: recent studies, such as https://arxiv.org/pdf/2402.19427, indicate that linear diagonal RNNs are memory bound, with computation dominated by memory transfers. Increasing the complexity of the sequential mechanism is thus a promising research area. Our paper shows that advancements towards efficient (perhaps sparse) non-diagonal computation are supported by increased expressivity. Our analysis also highlights specific components to target for enhancing compute with direct impacts on expressivity. Note that https://arxiv.org/pdf/2404.08819 arrives at similar conclusions, with drastically different tools but without identifying completely the hypothesis classes, as we instead do here.

---

### Decision · Program_Chairs · 2024-09-25

**Decision:**

Accept (poster)

**Comment:**

The paper presents a detailed study on the expressive power of State Space Models (SSMs) through the lens of Rough Path Theory, with a particular focus on the importance of the gating mechanism. The paper is well-structured and clearly written, making it accessible to readers who are familiar with the core concepts. The results are robust and provide valuable insights into the design choices made in past implementations of SSMs. These findings have the potential to guide the development of future variants of SSMs, contributing positively to the field.

However, there is some room for improvement. While the authors have made a commendable effort to include preliminary materials in the appendix, the paper may still be challenging for readers who are not well-versed in the required background. This could limit its accessibility to a broader audience. A more thorough explanation or simplification of these complex concepts within the main text would greatly enhance the paper's readability. Despite this, the overall quality of the work and its potential impact on future research in SSMs warrant its acceptance.